# Structures of autoinhibited and polymerized forms of CARD9 reveal mechanisms of CARD9 and CARD11 activation

Michael J. Holliday [1], Axel Witt[1], Alejandro Rodríguez Gama[2], Benjamin T. Walters [3], Christopher P. Arthur[4], Randal Halfmann[2,5], Alexis Rohou [4], Erin C. Dueber[1] & Wayne J. Fairbrother [1]

CARD9 and CARD11 drive immune cell activation by nucleating Bcl10 polymerization, but are held in an autoinhibited state prior to stimulation. Here, we elucidate the structural basis for this autoinhibition by determining the structure of a region of CARD9 that includes an extensive interface between its caspase recruitment domain (CARD) and coiled-coil domain. We demonstrate, for both CARD9 and CARD11, that disruption of this interface leads to hyperactivation in cells and to the formation of Bcl10-templating filaments in vitro, illuminating the mechanism of action of numerous oncogenic mutations of CARD11. These structural insights enable us to characterize two similar, yet distinct, mechanisms by which autoinhibition is relieved in the course of canonical CARD9 or CARD11 activation. We also dissect the molecular determinants of helical template assembly by solving the structure of the CARD9 filament. Taken together, these findings delineate the structural mechanisms of inhibition and activation within this protein family.

[1] Early Discovery Biochemistry Department, Genentech, South San Francisco, CA 94080, USA. [2] Stowers Institute for Medical Research, Kansas City, MO 64110, USA. [3] Biochemical and Cellular Pharmacology Department, Genentech, South San Francisco, CA 94080, USA. [4] Structural Biology Department, Genentech, South San Francisco, CA 94080, USA. [5] Department of Molecular and Integrative Physiology, University of Kansas Medical Center, Kansas City, KS 66160, USA. Correspondence and requests for materials should be addressed to A.R. (email: rohou.alexis@gene.com) or to E.C.D. (email: dueber.erin@gene.com) or to W.J.F. (email: fairbrother.wayne@gene.com)

aspase recruitment domain (CARD)-containing protein 9 (CARD9) and CARD-containing protein 11 (CARD11, aka. CARMA1) are paralogous proteins that act in a conserved manner as scaffolding proteins required to propagate signaling in immune cells[1]. CARD9 functions in myeloid cells during innate immune responses to activate NF-κB and p38 MAPK. The most well characterized triggers of CARD9 signaling are fungal carbohydrates, which interact with C-type lectin receptors and ultimately contribute to a Th17-type immune response critical for proper control of fungal infections[2–5]. CARD11 acts in lymphoid cells downstream of T- and B-cell receptors and also leads to activation of NF-κB[6,7]. In these cell types, CARD11 is critical for proper cellular activation and proliferation upon receptor engagement.

CARD9 and CARD11 share an N-terminal domain architecture comprising an N-terminal CARD followed by ~300 residues with high coiled-coil propensity, referred to as the coiled-coil domain. CARD11 additionally contains a C-terminal membrane-associated guanylate kinase (MAGUK) domain, as well as a linker domain between the coiled-coil and MAGUK domains known as the inhibitory domain (ID). Both CARD9 and CARD11 are thought to be held in autoinhibited states prior to activation, although the specific nature of these states remained uncharacterized prior to this study[8]. CARD11 is activated via phosphorylation of serine residues in the ID by protein kinase C, which interferes with a complex network of interactions between the ID, CARD, and coiled-coil domains required to maintain the autoinhibited state[9,10]. CARD9 activation has been reported by different groups to require either phosphorylation at T231 by PKCδ[11] or ubiquitination at Lys125 by TRIM62[12]. To our knowledge, no single study has investigated both of these modifications together, and, absent structural characterization of the system, no mechanistic explanation for CARD9 activation by either of them has been suggested.

Upon activation, both CARD9 and CARD11 recruit the same downstream binding partner, B-cell lymphoma/leukemia 10 (Bcl10), which interacts through its own CARD with the CARD of CARD9 or CARD11, a CARD–CARD interaction critical for subsequent signal propagation and NF-κB activation. The CARDs of CARD9 or CARD11 are thought to form a nucleating helical template that promotes polymerization of Bcl10 via the Bcl10 CARD, which, along with other domains of activated CARD9/11, then recruits downstream factors, including MALT1, cIAPs, TRAF6, and the LUBAC complex that mediate subsequent ubiquitination of multiple members of the complex, including both linear and K63-linked poly-ubiquitination of Bcl10[13–16]. These ubiquitination events ultimately lead to activation of IKK and subsequent phosphorylation, ubiquitination, and degradation of IκB, thereby releasing NF-κB to translocate to the nucleus and induce transcription. Regulation of the pathway at CARD9/11 is thus likely achieved by modulating assembly of the helical CARD template.

A number of disease-associated mutations have been identified in CARD9 and CARD11. In CARD9, genetic deletion in mice or loss-of-function mutations in humans lead to chronic fungal infections[17–21]. One intriguing CARD9 splice variant, identified as protective in inflammatory bowel disease, eliminates a TRIM62 binding site, preventing CARD9 signaling and bolstering the idea that TRIM62 ubiquitination of CARD9 is critical for its activation[12,22]. In CARD11, genetic deletion in mice or loss-of-function mutations in humans lead to immunodeficiency due to defects in activation and proliferation of both T- and B-cells[23,24]. A large number of mutations have also been identified in CARD11 associated with a range of lymphoproliferative disorders, including diffuse large B-cell lymphoma (DLBCL)[25,26]. Many of these mutations have been further characterized and

shown to exhibit constitutive CARD11 signaling and aberrant cell proliferation, confirming CARD11 as an oncogene[27–29]. Among these studies, the N-terminal CARD and coiled-coil domains emerged as mutational "hot-spots", leading Chan et al.[30] to conduct a high-throughput mutagenesis screen on the N-terminal 140 residues of CARD11, in the context of the full-length protein, in search of hyperactivating CARD11 mutants. This study identified 23 sites with hyperactivating mutations in CARD11, many of which mapped to a short stretch of residues (112–130) termed the "LATCH", that was shown to be critical in maintaining CARD11 in an autoinhibited state.

Here, we present a structure of the N-terminal region of CARD9, which exhibits an extensive autoinhibitory interface required to prevent constitutive activation in both CARD9 and CARD11. From this structural insight, we then define the distinct structural mechanisms of activation in CARD9 and CARD11 and demonstrate that, upon activation, both proteins form helical templates that directly nucleate Bcl10 polymerization. Finally, solving a Cryo-EM structure of the CARD9 CARD helical assembly, we structurally characterize the activated form of this protein family.

## Results

**An extensive CARD-coiled-coil interface in CARD9.** Both CARD9 and CARD11 are thought to be maintained in an autoinhibited state, but the structural basis for this inhibition had remained an open question. To probe the mechanism of inhibition, we expressed and purified $^{15}$N-labeled regions of both CARD9 and CARD11 comprising the CARD and the first predicted segment of coiled-coil that mediates homo-dimerization of each construct (CARD9$^{2–152}$ and CARD11$^{8–172}$, see Fig. 1a). Notably, CARD11$^{8–172}$ has been shown by Qiao et al.[15] to be sufficient to accelerate Bcl10 polymerization in vitro. While CARD11$^{8–172}$ remained stable and soluble at high concentrations, the construct exhibited a "molten globule" $^{15}$N-TROSY spectrum (Fig. 1b), indicative of significant conformational heterogeneity and/or self-association. While SEC-MALS analysis of CARD11$^{8–172}$ indicated a molecular weight of 36.5 kDa, consistent with an expected dimeric molecular weight of 39.4 kDa, dynamic light scattering analysis at 100 μM revealed a polydispersity index of 0.40, indicating some self-association, which was corroborated by a slight improvement in the nuclear magnetic resonance (NMR) spectral quality upon twofold dilution (Supplementary Fig. 1A). In contrast to CARD11$^{8–172}$, the CARD11 CARD alone exhibits a well-dispersed spectrum indicative of a single dominant conformational state[31].

Unlike CARD11$^{8–172}$, the homologous CARD9$^{2–152}$ construct (see Supplementary Fig. 1B for sequence alignment) exhibited a well-dispersed $^{15}$N-TROSY spectrum with fewer than 150 peaks (Fig. 1c, black), consistent with a single dominant conformation and minimal self-association. We found that, at the high concentrations required to collect three-dimensional (3D) NMR spectra for chemical shift assignment and structure determination efforts, CARD9$^{2–152}$ formed higher-order oligomers over the course of several hours, severely impacting NMR spectral quality (Supplementary Fig. 1C). We have shown previously that the CARD9 CARD is capable of forming an extensively domain-swapped dimer, and suspected that these oligomers were due to swap-dimer-mediated "daisy-chaining" of two or more coiled-coil-mediated dimers[31]. Zn$^{2+}$ binds to the CARD with sub-nanomolar affinity and slows interconversion of the monomer and domain-swapped CARD dimer ~50-fold, so we generated CARD9$^{2–152}$ complexed 1:1 with Zn$^{2+}$, which slowed oligomer formation sufficiently to allow for 3D NMR data collection on the dimer (Supplementary Fig. 1C).

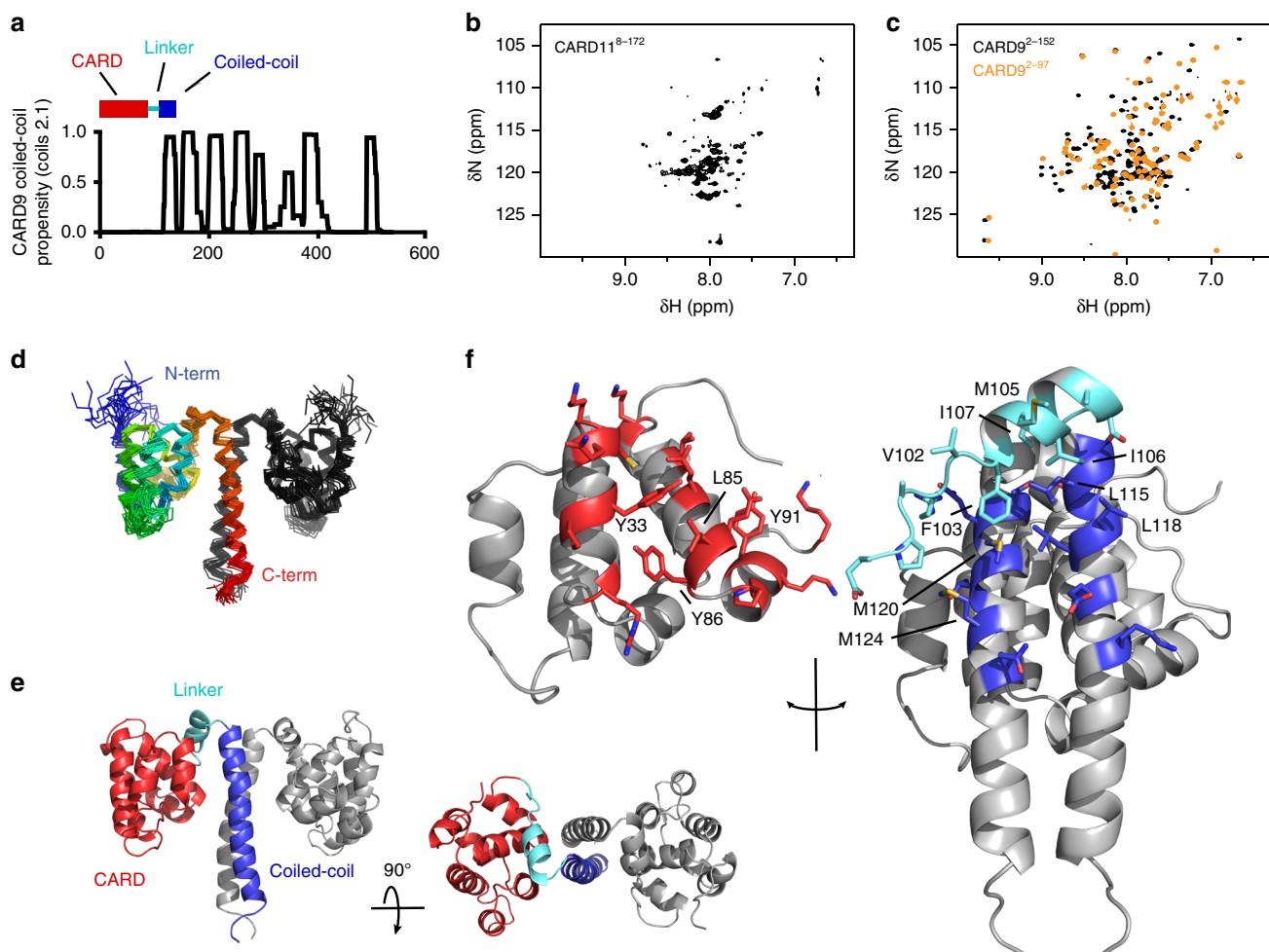

**Fig. 1** CARD9$^{2-142}$ forms a dimer with an extensive CARD-linker-coiled-coil interface. **a** Coiled-coil propensity for full-length CARD9, as predicted by the COILS server[73]. Schematic of the construct composition is shown above. **b** $^{15}$N-TROSY of CARD11$^{8-172}$, collected at 37 °C. **c** $^{15}$N-TROSY of the CARD9$^{2-152}$ dimer (black) and $^{15}$N-HSQC of CARD9$^{2-97}$ (orange) at 37 °C. The CARD9$^{2-97}$ spectrum was shifted to account for the difference in chemical shifts recorded by TROSY and HSQC experiments. **d** Ribbon diagram of the 20 lowest energy CARD$^{2-142}$ dimer structures calculated. One of the two monomers is colored as a rainbow from N- to C-terminus. **e** Lowest energy CARD9$^{2-142}$ structure, with sub-domains of one monomer colored as in Fig. 1a. **f** "Open-face" depiction of the CARD-coiled-coil interface. Residues of the CARD, linker, and coiled-coil for which heavy atoms are within 2.5 Å of another sub-domain in any of the lowest energy structures are shown as sticks and colored in red, cyan, and blue, respectively. Residues forming the hydrophobic core of the interface are noted

As shown in Fig. 1c, The CARD9$^{2-152}$ spectrum diverged significantly from that of the CARD9 CARD alone (CARD9$^{2-97}$), indicating that the CARD is not simply tethered to the coiled-coil, but instead makes extensive contact with the coiled-coil. After assigning the NMR backbone amide chemical shifts of CARD9$^{2-152}$, we found that large changes in amide peak positions map predominantly to one face of the CARD, containing α-helices 2, 5, and 6, suggesting these regions as the interaction interface (Supplementary Fig. 1D).

The ten C-terminal residues of CARD9$^{2-152}$ appear largely unstructured, and elimination of them minimally perturbed the $^{15}$N-TROSY spectrum. We thus purified and determined near-complete backbone and sidechain chemical shift assignments for the slightly shorter construct, CARD9$^{2-142}$, collected distance restraints in the form of non-specific and intermolecular-specific NOEs, collected residual dipolar coupling restraints, and determined the solution structure of the CARD9$^{2-142}$ dimer with a backbone RMSD among ordered residues of 1.1 Å (Fig. 1d, see Table 1 for statistics). While the overall backbone RMSD is 1.1 Å, alignment to a single CARD or the coiled-coil domain

alone yields backbone RMSDs of 0.7 Å and 0.8 Å, respectively, (Supplementary Fig. 1E).

The CARD9$^{2-142}$ structure consists of the N-terminal CARD, a linker (residues 98–111) comprising a short extended strand followed by a two-turn α-helix and a sharp turn at Gly111, and then 8 turns of α-helix. This final helical element forms a canonical dimeric coiled-coil interface consisting of Val and Leu residues. α-helices 2 and 5 of each CARD are packed against both the linker α-helix and both coiled-coil helices, with additional contacts made between the linker and CARD helices 5 and 6 (Fig. 1e, f). The CARD-linker-coiled-coil interface (hereafter referred to as the CARD-coiled-coil interface) is predominantly mediated through hydrophobic and aromatic packing, including Tyr33, Leu85, Tyr86, and Tyr91 of the CARD, Val102, Phe103, Met105, Ile106, and Ile107 of the linker, and Leu115, Leu118, Met120, and Met124 of the coiled-coil (Fig. 1f). There appear to be additional hydrogen bonding or electrostatic interactions mediated by Gln36, Glu81, Glu84, Arg101, Ser110, Gln117, and Glu122; however, these interactions are not sufficiently consistent among the structures determined to conclusively state which

**Table 1 Structural statistics for the CARD9$^{2-142}$ dimer**

| | |
|---|---|
| Assignments (%)[a] | 79 (94) |
| $^1$H | 72 (91) |
| $^{13}$C | 90 (96) |
| $^{15}$N | 78 (95) |
| NOE restraints | 3917 |
| Intra-residue [$i = j$] | 1102 |
| Sequential [$|i-j|$] = 1 | 998 |
| Medium range [$|i-j|$] < 5 | 770 |
| Long range [$|i-j|$] ≥ 5 | 764 |
| Intermolecular | 283 |
| Hydrogen bond constraints | 180 |
| Dihedral angle constraints | 428 |
| RDC restraints | 166 |
| Total number of restricting constraints | 4691 |
| Restricting constraints per restrained residue[b] | 16.9 |
| Long range [$|i-j|$] ≥ 5 | 3.8 |
| Total structures computed | 100 |
| Number of structures included | 20 |
| Distance violations per structure | |
| 0.1–0.2 Å | 47.15 |
| 0.2–0.5 Å | 10.15 |
| > 0.5 Å | 0 |
| R.m.s. of distance violation per constraint (Å) | 0.02 |
| Maximum distance violation (Å) | 0.45 |
| Dihedral angle violations per structure | |
| 1–10° | 17.3 |
| >10° | 0.3 |
| R.m.s. of dihedral angle violation per constraint (°) | 0.7 |
| Maximum dihedral angle violation (°) | 11.2 |
| R.m.s deviations[c] | |
| Backbone | 1.1 (2.0) |
| Heavy atoms | 1.4 (2.1) |
| Ramachandran[d] | |
| Most favored (%) | 90.7 |
| Additionally allowed (%) | 9.0 |
| Generously allowed (%) | 0.2 |
| Disallowed | 0.1 |

[a]Total assignment completeness, with backbone completeness reported in parentheses
[b]Residues 3–142 contain conformational restraining constraints
[c]Residues 10–142 reported, with all-residue RMSDs reported in parentheses
[d]Residues 10–142, calculated with Procheck

bonds are formed. Overall, among the 20 lowest energy structures calculated, on average 870 Å$^2$ are buried for each CARD, comprising ~15% of the total solvent exposed surface area of each CARD.

**An interface critical for CARD9 and CARD11 autoinhibition.**
Amino acids 5–131 of CARD9 share 51% identity with residues 17-143 of CARD11, with no gaps in alignment (Supplementary Fig. 1B). The CARD9 residues homologous to the "LATCH" region required for CARD11 autoinhibition comprise much of the CARD-interacting residues in the linker and coiled-coil (Supplementary Fig. 1F)[30]. We therefore mapped the known disease-associated or mutagenesis-identified hyperactivating sites in this region of CARD11 onto the CARD9$^{2-142}$ structure[25–29]. As shown in Figs 2a, b, all oncogenic and disease-associated mutations, as well as 20 of the 23 sites identified in the random mutagenesis screen by Chan et al.[30], map to the CARD-coiled-coil interface. This finding suggests that the interface is crucial for autoinhibition of CARD11.

Given the degree of homology with CARD11 and extent of the interface in CARD9$^{2-142}$, we suspected that the CARD-coiled-coil interface acts in an autoinhibitory manner in CARD9 as well. We generated a panel of homologous mutants in full-length CARD9 and transiently transfected them into HEK293 cells

containing an NF-κB-inducible reporter gene. As shown in Fig. 2c, overexpression of WT CARD9 weakly induced NF-κB activation as compared to CARD11 lacking its inhibitory domain (CARD11-ΔID), which robustly activated NF-κB, consistent with previous studies[8,32]. Six of the 11 CARD-coiled-coil-disrupting mutants tested consistently and robustly activated NF-κB by 4–12-fold, demonstrating that disruption of the CARD-coiled-coil interface is sufficient to activate both CARD9 and CARD11. Four of the mutants tested (C37Y, E81D, L85Y, and Y86F) appear to destabilize CARD9, indicated by significantly lower protein expression; increasing the concentration of transfected plasmid had a minimal effect on protein levels of these mutants. For L85Y, we observed consistently higher NF-κB activation despite the lowered protein expression, confirming that the L85Y mutation is activating. As we were unable to match WT protein expression for C37Y, E81D, and Y86F, we were unable to determine conclusively whether these mutations activate CARD9 as they do CARD11.

Closer examination of the residues involved in activating mutations (Fig. 2d), reveals the likely mechanisms through which they disrupt the interface. L85, F103, I107, and L115 all pack tightly together in the hydrophobic CARD-coiled-coil interface; disruption of this packing would likewise disrupt this tight interaction. G111 acts as a hinge between the linker and coil-coil, adopting Ramachandran angles (ϕ 81–139°, ψ 155–173°) incompatible with any non-glycine residue. Introduction of a non-glycine residue here likely disrupts this tight turn, preventing the linker from properly packing against the CARD and coiled-coil. G114 adopts the standard Ramachandran angles of an α-helix, but the tight turn at G111 brings the linker α-helix in proximity to G114, such that introduction of any side chain would likewise disrupt the packing of the linker. In any of these cases, weakening the CARD-coiled-coil interaction presumably frees the CARD to sample an open conformation, allowing for homo- and heterotypic interactions required for signal propagation, explored further below.

**A weak CARD11 autoinhibitory interface reinforced by the ID.**
We reasoned that the increased conformational heterogeneity of CARD11$^{8-172}$, relative to CARD9$^{2-152}$ (Fig. 1b, c), may result from a more labile CARD-coiled-coil interface. Additional interactions with the ID would therefore be required to maintain the closed, autoinhibited state. To test whether the CARD11 CARD-coiled-coil interface is less functionally inhibitory than that of CARD9, we generated a chimeric expression construct in which we replaced the homologous N-terminal residues comprising the CARD, linker, and CARD-interacting stretch of coiled-coil of CARD9 with that of CARD11 (CARD11$^{1-143}$/CARD9, schematic shown in Fig. 3a, sequence alignment shown in Supplementary Fig. 1B). Upon overexpression in HEK293 cells, we found that the chimeric CARD11$^{1-143}$/CARD9 construct induced robust NF-κB activation comparable to levels of activation seen in CARD-coiled-coil-disrupting mutants (Fig. 2c) and well above that observed for CARD9$^{WT}$, even at much higher CARD9$^{WT}$ protein levels (Fig. 3b). These data indicate that the CARD11 CARD-coiled-coil interface is less inhibitory on its own than that of CARD9.

To test whether the function of the CARD11 ID is to maintain a closed CARD-coiled-coil conformation, we generated the complementary chimera in which the N-terminal residues comprising the CARD, linker, and CARD-interacting region of the coiled-coil of CARD11-ΔID were replaced with the homologous residues in CARD9 (CARD9$^{1-131}$/CARD11-ΔID). CARD11-ΔID induced high levels of NF-κB activation, consistent with previous studies[8,32]. However, CARD9$^{1-131}$/CARD11-ΔID,

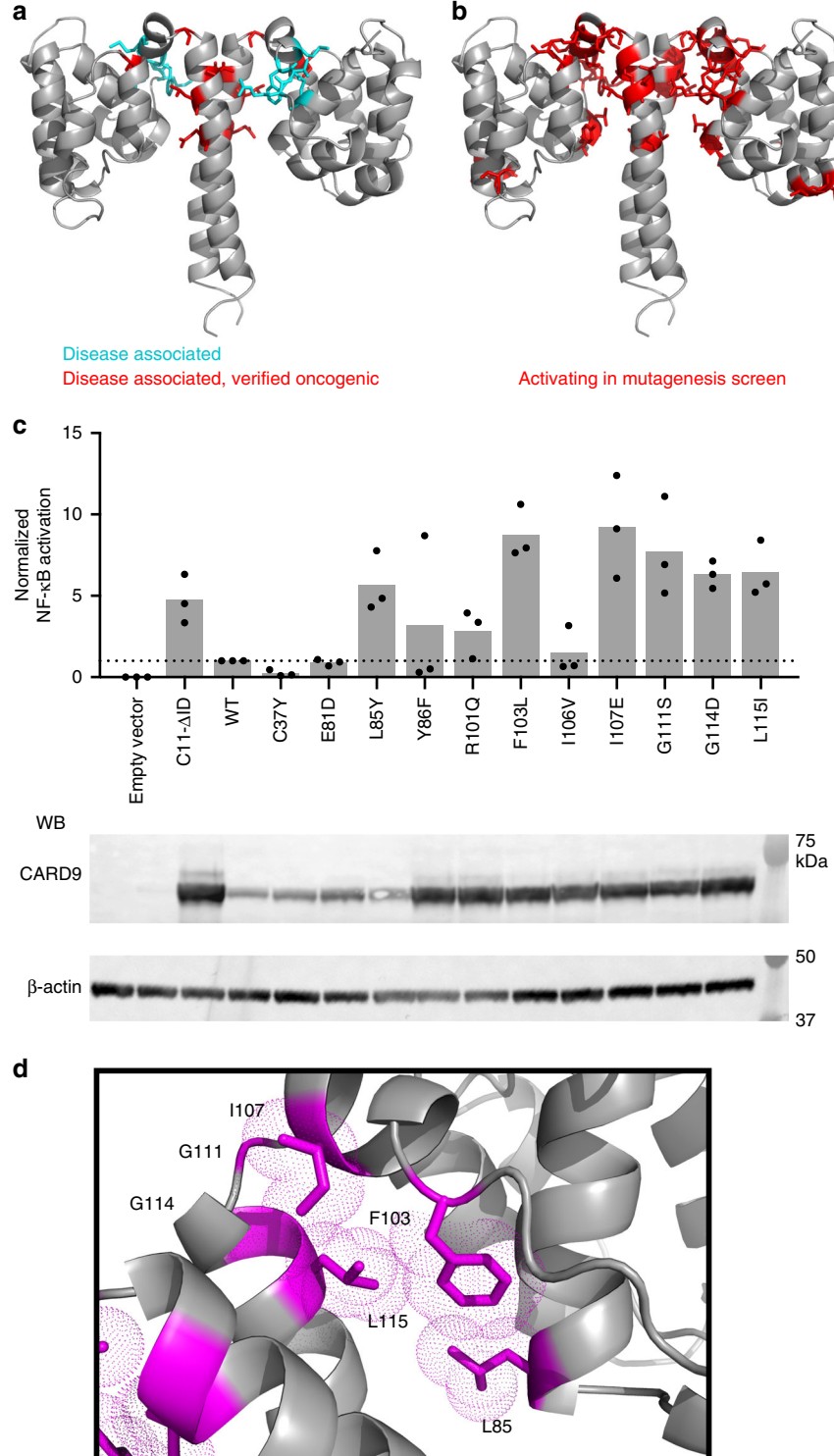

**Fig. 2** The CARD-coiled-coil interface is critical for maintenance of an autoinhibited state. **a** Disease associated mutants in CARD11 (cyan) and disease associated mutants for which follow-up work confirmed that the mutants were oncogenic (red), mapped onto the homologous residues of CARD9 in the lowest energy NMR structure. **b** Hyperactivating CARD11 mutants identified by Chan et al.[30] in a randomized mutagenesis screen, mapped onto the homologous residues of CARD9[2−142]. **c** Normalized NF-κB activation in HEK-293 cells upon overexpression of WT or mutant full-length CARD9. The known activator CARD11-ΔID was used as a positive control. Three biologically independent experiments were undertaken, the outcomes of which are shown as dots. The mean value for each construct is depicted as a gray bar. Dotted line represents the value of the WT construct, to which the other values were normalized. Representative western blot is shown for CARD9, demonstrating the total protein expression levels of the constructs. Molecular weights of protein standards are indicated. Source data are provided in Source Data file. **d** The CARD9 CARD-coiled-coil interface, with residues for which mutation consistently activates NF-κB signaling depicted in magenta; sidechain heavy atom van der Walls radii are depicted as dotted surfaces

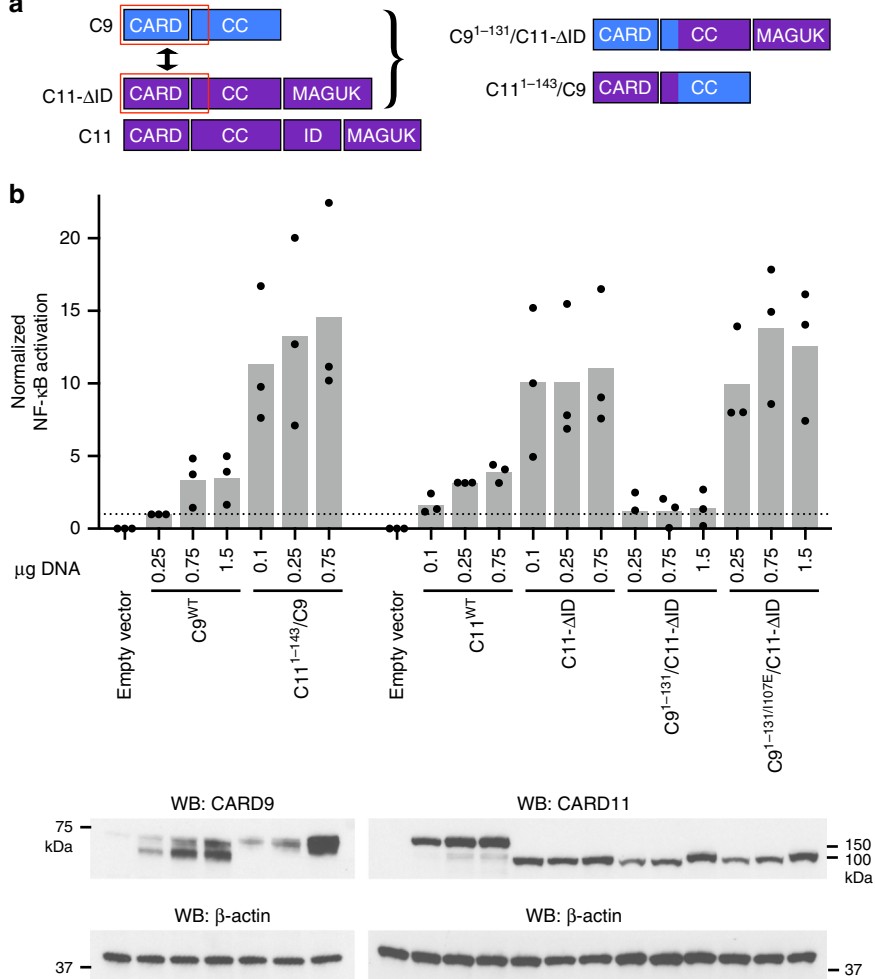

**Fig. 3** The CARD11 CARD-coiled-coil interface is weaker than that of CARD9, but is reinforced by the ID. **a** Cartoon schematic of the domain architecture of CARD9, CARD11-ΔID, and CARD11 along with the chimeric constructs CARD9[1−131]/CARD11-ΔID and CARD11[1−143]/CARD9, wherein the N-terminal residues comprising the CARD, linker, and CARD-interacting stretch of the coil-coil are swapped between the two proteins (See Supplementary Fig. 1B). **b** Normalized NF-κB activation in HEK-293 cells upon overexpression of the indicated constructs. Three biologically independent experiments were undertaken, the outcomes of which are shown as dots. The mean value for each construct is depicted as a gray bar. Dotted line represents the value of the CARD9[WT] construct, to which the other values were normalized. Representative western blot is shown for CARD9 and CARD11 constructs, using antibodies specific for regions outside of the N-terminal swapped domain or ID. Source data are provided in Source Data file

which introduces a more robust CARD-coiled-coil interface, induced minimal NF-κB activation (Fig. 3b). To confirm that CARD9[1−131]/CARD11-ΔID retains competency to signal, we also generated a chimeric construct that additionally contained a CARD-coiled-coil-disrupting mutation (CARD9[1−131/I107E]/CARD11-ΔID); as shown in Fig. 3b, this construct activated NF-κB to a level comparable to CARD11-ΔID, indicating that the weak signaling induction by CARD9[1−131]/CARD11-ΔID is due to autoinhibition by the tightly closed CARD-coil-coil interface of CARD9.

**CARD9 ubiquitination disrupts the autoinhibitory interface.** CARD9 has no known region analogous to the ID, and based on the weak NF-κB activation induced by the CARD9[1−131]/CARD11-ΔID chimeric construct (Fig. 3b), the CARD9 CARD-coiled-coil interface is sufficiently tight alone to maintain the CARD in an inhibited state. These observations suggest that signaling-induced disruption of the autoinhibitory CARD-coiled-coil interface must proceed through a different mechanism for CARD9 than for CARD11. Cao et al.[12] reported that CARD9 is specifically ubiquitinated at Lys125 by TRIM62 and that this

ubiquitination is required for signal propagation through CARD9. The dimeric CARD9[2−142] structure shows that Lys125 is not directly involved in the CARD-coiled-coil interface; however, Lys125 is in proximity to the CARD, suggesting that ubiquitin conjugation at Lys125 could sterically disrupt the CARD-coiled-coil interface.

To generate sufficient quantities of ubiquitinated CARD9[2−152], we utilized a chemical ligation approach. A CARD9[2−152] construct was generated with both cysteines (C10 and C37) mutated to serine, and Lys125 mutated to cysteine (CARD9[2−152/SSC]). In parallel, a ubiquitin variant was generated for which the C-terminal glycine was mutated to cysteine (Ub[G76C]). After individually purifying each component, the two proteins were conjugated using the short bi-functional maleimide crosslinker bismaleimidoethane (BMOE) and purified by size exclusion chromatography, yielding CARD9[2−152/SSC] dimers with ~95% ubiquitin conjugation (Supplementary Fig. 2A). As CARD9[2−152/SSC] and Ub[G76C] were purified separately, we generated ligated preparations for which only CARD9[2−152/SSC] or Ub[G76C] was [15]N-labeled, allowing selective investigation of the impact of ligation on each of the proteins by NMR.

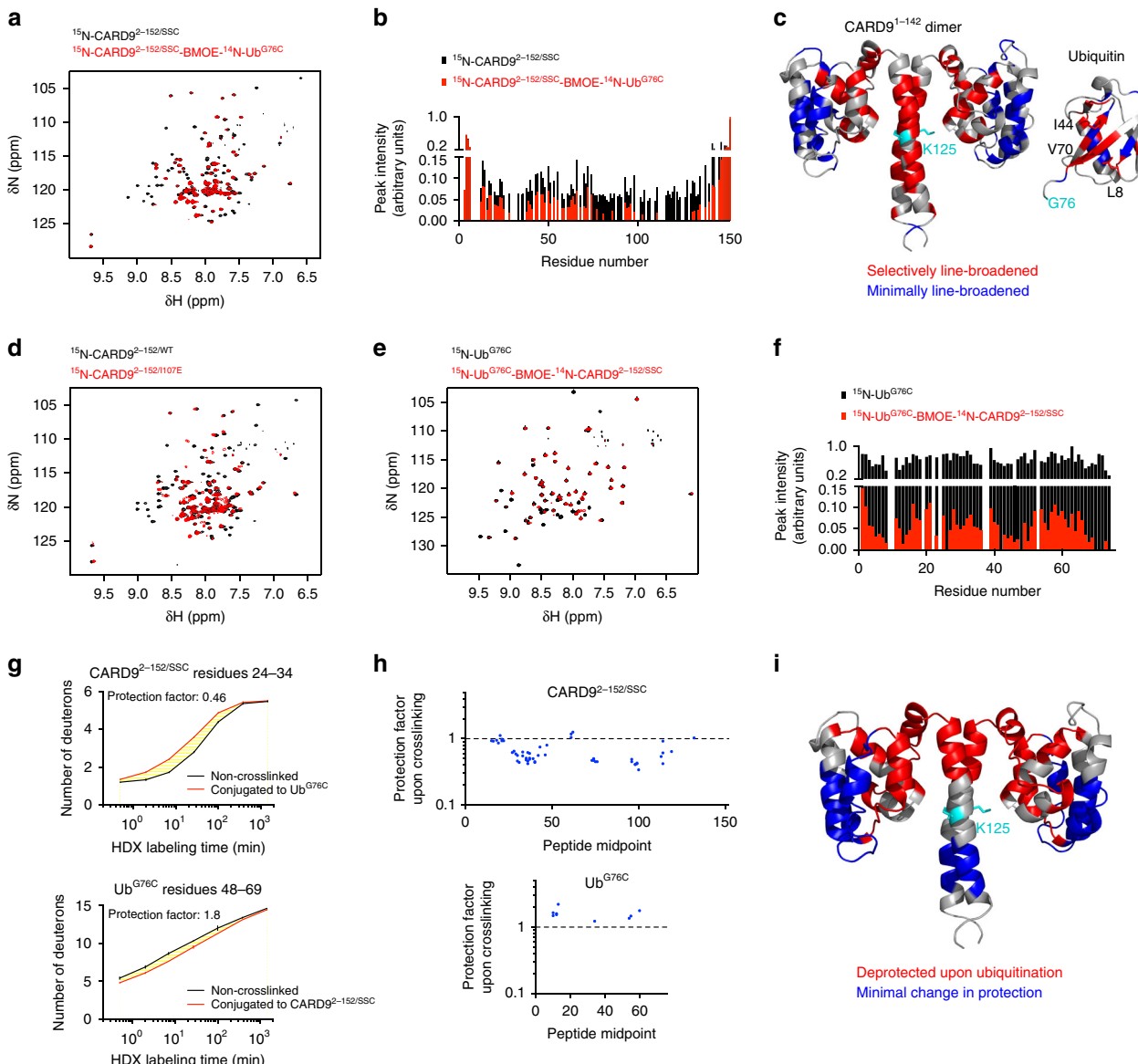

**Fig. 4** Ubiquitination of CARD9 at Lys125 disrupts the CARD-coiled-coil interface. **a** $^{15}$N-TROSYs of CARD9$^{2-152/SSC}$ apo (black) or conjugated to $^{14}$N-Ub$^{G76C}$ (red). **b** Peak intensities of spectra in **a** for all assigned, non-overlapped peaks. **c** Lowest energy CARD9$^{2-142}$ dimer structure (left) and ubiquitin crystal structure (PDBID 1UBQ [http://dx.doi.org/10.2210/pdb1UBQ/pdb], right). Selectively line-broadened residues (<25% apo intensity for CARD9$^{2-142}$ and <5% apo intensity for ubiquitin) are colored red. Minimally line-broadened residues (>50% apo intensity for CARD9$^{2-142}$ and >10% apo intensity for ubiquitin) are colored blue. Residues Lys125 in CARD9$^{2-142}$ and Gly76 in ubiquitin are colored cyan. **d** $^{15}$N-TROSYs of CARD9$^{2-152/WT}$ (black) and CARD9$^{2-152/I107E}$ (red). **e** $^{15}$N-TROSYs of Ub$^{G76C}$ apo (black) or conjugated to $^{14}$N-CARD9$^{2-152/SSC}$ (red). **f** Peak intensities of spectra in panel **e** for all non-overlapped peaks. **g** Deuterium exchange as a function of labeling time for representative peptides from CARD9$^{2-152/SSC}$ and Ub$^{G76C}$ unconjugated (black) or conjugated (red). **h** Global protection factors for CARD9$^{2-152/SSC}$ and Ub$^{G76C}$ after conjugation as compared to apo. Each dot represents the mid-point of a single identified peptide. Dotted line indicates a protection factor of 1, indicating no change upon conjugation. Values less than one indicate de-protection, i.e., faster deuterium exchange. **i** CARD9$^{2-142}$ dimer structure with regions de-protected upon ubiquitination plotted in red and unaffected regions plotted in blue. Lys125 is indicated in cyan. All source data are provided in Source Data file

The overall peak dispersion and linewidths remain comparable between CARD9$^{2-152/SSC}$ and CARD9$^{2-152}$ (Supplementary Fig. 2B), indicating that the mutations have not disrupted the CARD-coiled-coil interface, confirming that Lys125 is not crucial for maintenance of the interface, and allowing us to assign 79% of the backbone amide chemical shifts in CARD9$^{2-152/SSC}$. Mixing $^{14}$N-Ub$^{G76C}$ and $^{15}$N-CARD9$^{2-152/SSC}$ in a 1:1 ratio, we observed no changes in the CARD9$^{2-152/SSC}$ spectrum, indicating that, in the absence of conjugation, the two proteins do not appreciably interact (Supplementary Fig. 2C). However, as shown in Fig. 4a,

the BMOE-conjugated $^{15}$N-CARD9$^{2-152/SSC}$—$^{14}$N-Ub$^{G76C}$ spectrum exhibited both a global line broadening, attributable to the larger molecular weight of the complex (52.1 kDa vs. 34.6 kDa), as well as site-specific, selective line broadening indicative of intermediate time-scale dynamics for certain regions of CARD9$^{2-152/SSC}$ (Fig. 4b). Mapping all residues with a >75% reduction in peak intensity onto the CARD9$^{2-142}$ structure (Fig. 4c, red), we found that the selectively line-broadened residues localize predominantly to the CARD-coiled-coil interface, while unperturbed residues (<50% reduction, Fig. 4c, blue)

are distal from the interface. We compared the $^{15}$N-CARD9$^{2-152/}$$^{SSC}$—$^{14}$N-Ub$^{G76C}$ spectrum to those of the constructs containing the activating mutations I107E (CARD9$^{2-152/I107E}$) or L115I (CARD9$^{2-152/L115I}$) and found that the activating mutants induce selective line broadening to a similar extent and to nearly identical residues (Fig. 4d and Supplementary Fig. 2D), suggesting that this pattern of line broadening represents a signature of functionally significant CARD-coiled-coil disruption.

The inversely labeled $^{14}$N-CARD9$^{2-152/SSC}$—$^{15}$N-Ub$^{G76C}$ complex likewise exhibited both global and selective line broadening as compared to $^{15}$N-Ub$^{G76C}$ alone (Fig. 4e, f). The degree of global line broadening is more severe in this case due to the much larger molecular weight difference (52.1 kDa vs. 8.8 kDa). Highlighting those residues with the greatest peak intensity reduction (>95%, Fig. 4c, red) onto a ubiquitin crystal structure, the line broadened residues map predominantly to a well characterized hydrophobic patch that includes L8, I44, and V70—the most common protein–protein interaction interface on ubiquitin[33].

The selective line broadening in both CARD9 and ubiquitin suggests that the hydrophobic patch of ubiquitin competes with the CARD to interact with the hydrophobic surface of the coiled-coil and/or linker region and thus displaces the CARD. To test whether the CARD is physically displaced by ubiquitin conjugation, we monitored conjugated and unconjugated CARD9$^{2-152/SSC}$ and Ub$^{G76C}$ by time-resolved mass spectrometry-based hydrogen-deuterium exchange (HDX). By comparing deuterium incorporation over time between a conjugated and unconjugated sample, a protection factor can be assigned to each peptide identified in both CARD9$^{2-152/SSC}$ and Ub$^{G76C}$ (representative examples shown in Fig. 4g, see Source Data for all identified peptides); protection factors of less than one correspond to de-protection (i.e., faster exchange, structural destabilization) upon ubiquitination, while protection factors of greater than one correspond to protection. Globally, we identified several regions of CARD9$^{2-152/SSC}$ that exhibit de-protection upon ubiquitination (Fig. 4h), consistent with the displacement of the CARD and elimination of structurally stabilizing interactions. By comparison of many overlapping peptides, we found that residues 24–38, 71–78, 83–85, 93–102, 106–108, and 116–120 were de-protected upon ubiquitination, while residues 4–23, 39–66, and 129–135 remained unaffected. Mapping these residues onto the structure, we found increases in exchange rates along the CARD-coiled-coil interface upon ubiquitination, while residues distal from the interface remained unaffected (Fig. 4i), corroborating our interpretation of the NMR data that Lys125 ubiquitination physically displaced the CARD. Ub$^{G76C}$ exhibited a modest global protection upon conjugation, consistent with a stabilizing interaction between the conjugated ubiquitin and CARD9$^{2-152/SSC}$. Finally, we compared HDX rates between CARD9$^{2-152/WT}$ and a construct containing an activating mutation, CARD9$^{2-152/I107E}$. At both the individual peptide level (Supplementary Fig. 2E) and globally (Supplementary Fig. 2F), ubiquitination of CARD9$^{2-152/SSC}$ induced a similar pattern and extent of de-protection as the I107E mutation, indicating that ubiquitination at Lys125 displaces the CARD to a functionally significant degree.

## CARD9 CARD-coiled-coil disruption promotes helical assembly.

Canonically, death domains like CARDs propagate signaling cascades via homo- or heterotypic interactions with other death domains, forming helical assemblies[34]. For CARD9 and CARD11, the CARD–CARD interaction with Bcl10 is critical for signal propagation, and CARD11$^{8-172}$ has been shown to accelerate polymerization of Bcl10 filaments in vitro[15]. We hypothesized that the CARD-coiled-coil interaction sequesters the CARD9 and CARD11 CARDs in the absence of activation, thus preventing homotypic interactions required to generate a Bcl10-nucleating template.

To test this hypothesis, we purified recombinantly expressed CARD9$^{2-152}$ with mutations identified as NF-κB activating (Fig. 2b). We found that CARD9$^{2-152/I107E}$ spontaneously formed filaments after a short incubation at elevated temperatures, which are readily visible by negative-stain electron microscopy (NS-EM, Fig. 5a). As we have shown previously for the CARD9 CARD alone, filament formation was hindered in the presence of zinc. Upon addition of excess EDTA to 0.5 mM CARD9$^{2-152/I107E}$ that had been purified with a 1:1 concentration of zinc, filaments were readily observable by NS-EM after 10 min at 25 °C, while none were observed without adding EDTA (Supplementary Fig. 3A). In the presence of zinc, filaments could be induced by elevating the temperature to 37 °C, suggesting that zinc inhibits but does not completely block filament formation. A second mutant, CARD9$^{2-152/L115I}$, was additionally identified to form filaments in vitro that are similarly modulated by Zn$^{2+}$. The L115I mutation appears to be somewhat less permissive of filament formation, however, as filaments formed within 10 min upon Zn$^{2+}$ removal only at 37 °C, and not at 25 °C. Under none of these conditions were filaments of CARD9$^{2-152/WT}$ ever found, even after removal of Zn$^{2+}$ from 1 mM CARD9$^{2-152/WT}$ followed by incubation for 16 h at 37 °C (Fig. 5a).

To test whether the CARD-coiled-coil interaction blocks CARD9 CARD assembly in a cellular context, we utilized Distributed Amphifluoric FRET (DAmFRET)[35]. Briefly, in this method, a protein of interest is tagged with the photoconvertible fluorescent protein mEos3.1 and expressed over a range of concentrations in cells. By exposing the cells to a limiting dose of 405 nm light, a fraction of the fluorophore molecules are photoconverted, allowing them to act as FRET acceptors to the fluorophore molecules that have not converted. This system thus allows for monitoring protein self-assembly, which increases FRET signal, as a function of the protein's concentration in cells. The DAmFRET assay has been established in the budding yeast, S. cerevisiae, which has the advantage of not containing native death domain proteins or associated regulatory mechanisms that could potentially obscure intrinsic CARD assembly properties. For the CARD9 CARD alone (CARD$^{1-97}$), DAmFRET revealed a sharp increase in polymerization at concentrations above ~100 μM (Fig. 5b and Supplementary Fig. 3B, C), consistent with previous in vitro measurements[31]. Inclusion of the coiled-coil (CARD9$^{1-142}$) blocked this polymerization while simultaneously increasing the FRET level of soluble protein (Fig. 5b, inset), consistent with the expected formation of dimers. Disrupting the CARD-coiled-coil interface within this construct via mutation (CARD9$^{1-142/I107E}$) restored polymerization at high concentrations (Fig. 5b and Supplementary Fig. 3B, C). Intriguingly the mutation also increased FRET levels for the soluble protein (Fig. 5b, inset), suggesting an increase in oligomerization even at subsaturating concentrations. The CARD9 CARD-coiled-coil interaction thus inhibits CARD9 CARD polymerization in living cells, as well as in vitro.

## CARD9 templates Bcl10 nucleation in vitro and in cells.

We anticipated that the observed in vitro filaments (Fig. 5a) likely represent the nucleating template that seeds Bcl10-filament formation. To test this hypothesis, we utilized a fluorescence polarization (FP)-based Bcl10 polymerization assay described previously[31], based on the assay developed by Qiao et al.[15]. Bcl10 was purified with an N-terminal MBP tag linked by a TEV protease cleavage site and sparsely labeled with a fluorophore. The MBP tag blocks Bcl10 polymerization; however, the MBP tag

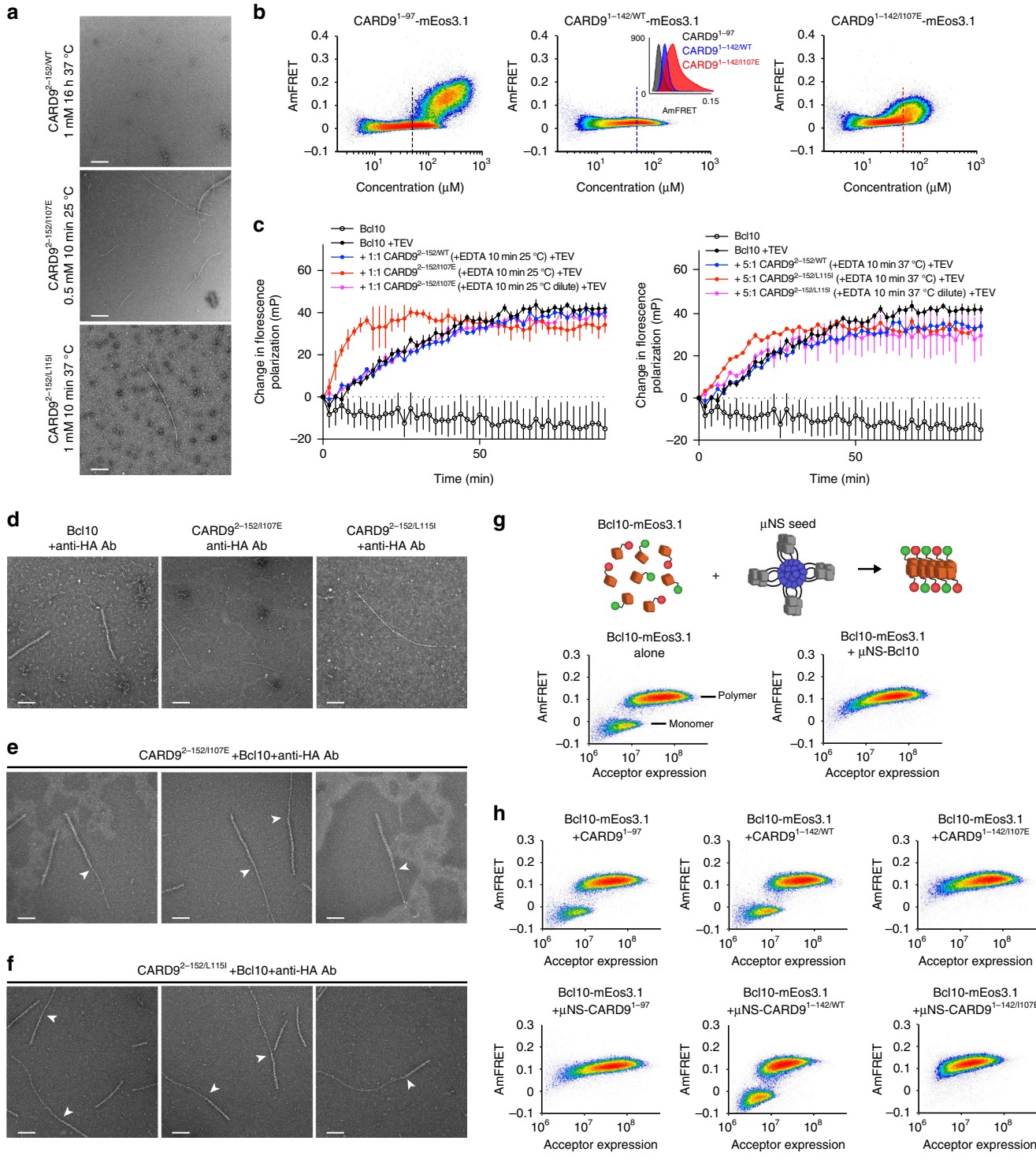

is rapidly removed upon addition of TEV protease, with >50% cleaved in 2 min (Supplementary Fig. 3D). Upon removal of MBP, Bcl10 spontaneously forms filaments, leading to an increase in FP. Factors that nucleate Bcl10 induce a more rapid increase in FP upon addition of the protease.

Utilizing this FP assay, we found that CARD9$^{2-152/I107E}$ and CARD9$^{2-152/L115I}$ filaments accelerate Bcl10 polymerization, while CARD9$^{2-152/WT}$, or mutant constructs under conditions that do not generate filaments, fail to accelerate Bcl10 polymerization. Specifically, we added excess EDTA to 1:1 Zn$^{2+}$-complexed CARD9$^{2-152/I107E}$ and incubated it at either a

concentration that allows (0.5 mM) or does not allow (4 μM) filament formation. Adding an equimolar concentration of the filamentous preparation to 2 μM MBP-Bcl10, but not the non-filamentous preparation, accelerated Bcl10 polymerization (Fig. 5c). As shown in Supplementary Fig. 3E, Zn$^{2+}$-bound CARD9$^{2-152/I107E}$ is also capable of promoting Bcl10 polymerization, but only when first incubated under conditions permissive of filaments. Under no conditions was the CARD9$^{2-152/WT}$ construct able to accelerate Bcl10 polymerization. CARD9$^{2-152/L115I}$ is likewise able to promote Bcl10 polymerization only under conditions for which it forms

**Fig. 5** CARD-coiled-coil disruption promotes Bcl10-nucleating CARD9 helical assembly. **a** NS-EM micrographs of CARD9$^{2-152/WT}$, CARD9$^{2-152/I107E}$, and CARD9$^{2-152/L115I}$ after removal of Zn$^{2+}$ by EDTA and incubation at the indicated concentration, time, and temperature. Incubation of CARD9$^{2-152/WT}$ for shorter times, at lower temperature (25 °C) or lower concentration (0.5 mM) also failed to produce filaments by NS-EM. Scale bars of all panels are 200 nm. **b** DAmFRET assay performed on CARD9$^{1-97}$, CARD9$^{1-142/WT}$, or CARD9$^{1-142/I107E}$. Inset depicts AmFRET binned at 50–60 μM for each construct, with 50 μM indicated by vertical dashed lines. See Supplementary Fig. 4D for replicate data, statistical analysis, and data binned at 100–200 μM. **c** Bcl10 FP nucleation assay. MBP-Bcl10 was mixed with CARD9$^{2-152}$ preparations as indicated. EDTA addition and incubation of CARD9$^{2-152}$ constructs occurred prior to initiation of nucleation assay; 1:1 and 5:1 refers to the molar ratio during the FP assay, with the final concentration of Bcl10 always set to 2 μM. TEV protease was added at $t = 0$ to indicated samples. Vertical bars represent the s.d. of three technical replicates. The data shown in the two plots were collected in the same experiment and were plotted separately to aid in visualization; hence, the 'Bcl10' and 'Bcl10 + TEV' data are identical in both plots. Source data are provided in Source Data file. **d** Bcl10, CARD9$^{2-152/I107E}$, or CARD9$^{2-152/L115I}$ filaments, 2 min after addition of TEV protease and with addition of anti-HA antibody during NS-EM grid preparation. **e, f** Direct templating of Bcl10 by CARD9$^{2-152/I107E}$ (**e**) or CARD9$^{2-152/L115I}$ (**f**) filaments. Grids were prepared 2 min after addition of TEV protease and with anti-HA antibody added during NS-EM grid preparation. White arrows indicate CARD9$^{2-152}$-to-Bcl10-filament transitions. For panels **d–f**, samples were prepared under identical conditions and concentrations as in **c** with all grids prepared 2 min after addition of TEV protease. **g** Schematic of genetically encoded seeds. DAmFRET assay for Bcl10-mEos3.1 alone (left), showing that it partitions cells into overlapping populations that contain either monomer alone or polymer; or for Bcl10-mEos3.1 with co-expressed with μNS-Bcl10 (right), which shifts all cells to the polymer-containing population. **h** DAmFRET assay for Bcl10-mEos3.1 with co-expressed CARD9$^{1-97}$, CARD9$^{1-142/WT}$, or CARD9$^{1-142/I107E}$ either alone or tethered to μNS

filaments (Fig. 5c). These findings indicate that the CARD9$^{2-152}$ helical assembly, and not the dimeric form of the protein, is capable of accelerating Bcl10 polymerization and that disruption of the CARD-coiled-coil interface promotes this activity.

We have previously shown that CARD9$^{2-97}$ helical assemblies are capable of directly templating Bcl10 polymerization. To demonstrate that the CARD9$^{2-152/I107E}$ and CARD9$^{2-152/L115I}$ filaments are likewise accelerating Bcl10 polymerization through direct templating, we visualized, by NS-EM, CARD9$^{2-152}$ filament-nucleated Bcl10 filaments shortly after addition of TEV. To distinguish CARD9$^{2-152}$ and Bcl10 filaments, we utilized a Bcl10 construct with a C-terminal HA tag and added anti-HA antibody after filament formation to specifically decorate Bcl10 filaments (Fig. 5d). After adding anti-HA antibody to CARD9$^{2-152/I107E}$- or CARD9$^{2-152/L115I}$-nucleated Bcl10 filaments, we visualized large numbers of heterotypic filaments, distinguishable by one thin, "sharp" end (CARD9$^{2-152}$) and one thicker, "fuzzy" end (Bcl10), demonstrating direct templating by the CARD9$^{2-152}$ filaments (Fig. 5e, f). As was found for nucleation by CARD-only filaments, we never observed more than a single CARD9-to-Bcl10 transition in a given filament, consistent with unidirectional Bcl10 polymerization[31,36].

For CARD9 to template Bcl10 nucleation in vivo, Bcl10 must have a large enough kinetic barrier to nucleation that the unassembled protein is supersaturated in cells. We used DAmFRET to test for such a barrier. As shown in Fig. 5g, cells expressing Bcl10-mEos3.1 alone partitioned into both low- and high-FRET populations, indicative of soluble and polymerized Bcl10, respectively. These two populations of cells had discrete FRET values and occurred at overlapping concentrations of protein, indicating that the initial formation of polymers within cells is indeed rate-limited by nucleation. To confirm this interpretation, we expressed genetically encoded seeds of Bcl10 in trans. These seeds consist of Bcl10 conjugated to μNS, a virus protein that forms dynamic condensates that sequester the conjugated protein to high local concentration[37]. Co-expressing Bcl10-μNS shifted the entire low FRET Bcl10-mEos3.1 population of cells to the high-FRET population, verifying that the soluble state of Bcl10-mEos3.1 in the low FRET cells is supersaturated (Fig. 5g).

We next tested if the CARD9 CARD (CARD9$^{1-97}$) can likewise nucleate Bcl10-mEos3.1 and, indeed, it did (Fig. 5h, left). This ability depended on its fusion to μNS, consistent with our in vitro observation that the CARD9 CARD could only accelerate Bcl10 polymerization when it was itself in a multimeric state[31]. Finally, we asked if the interaction of CARD with the coiled-coil

inhibits this activity. As expected, CARD9$^{1-142/WT}$ failed to nucleate Bcl10, even when it was expressed as a fusion to μNS (Fig. 5h, middle). Given that the CARD9$^{1-97}$-μNS fusion robustly nucleated Bcl10, this result confirms that the coiled-coil region inhibits CARD9 activity along with inhibiting polymerization. Conversely, disruption of the CARD-coiled-coil interaction (CARD9$^{1-142/I107E}$) resulted in complete nucleation of Bcl10 (Fig. 5h, right). Remarkably, this activity did not require μNS conjugation, suggesting that the presence of the coiled-coil along with disruption of the CARD-coiled-coil interface allows CARD9$^{1-142/I107E}$ to form helical templates on its own.

## CARD11$^{8-172}$ also forms Bcl10-nucleating helical templates.
Consistent with a more labile CARD-coiled-coil interface in CARD11 (Figs. 1b and 3), CARD11$^{8-172}$ was able to polymerize in vitro without introduction of a mutation. When incubated at 37 °C, solutions of 1 mM CARD11$^{8-172}$, but not 4 μM CARD11$^{8-172}$, were observed to contain filaments as visualized by NS-EM (Supplementary Fig. 4A). The filament-containing sample of CARD11$^{8-172}$ strongly accelerated Bcl10 polymerization, whereas the sample that lacked filaments had no effect (Supplementary Fig. 4B). Although CARD9$^{2-152}$ filament-containing samples formed intractable gels after extended incubation, CARD11$^{8-172}$ filament-containing samples instead remained clear. We thus utilized SEC to fractionate the filament-containing sample, yielding three distinct species: dimers, filaments, and fairly uniform oligomers of ~300 kDa (Supplementary Fig. 4C, D). Separation was incomplete, however, as small numbers of large filaments could be identified by NS-EM in the dimer and oligomer fractions (Supplementary Fig. 4E); presumably these filaments were retained by the in-line pre-column filter and subsequently dislodged over the course of the SEC run. At a 1:5 molar ratio, the filament fraction was capable of robust Bcl10 nucleation, while the dimeric and oligomeric fractions induced nucleation only slightly above background (Supplementary Fig. 4F), indicating that, as with CARD9, the CARD11 helical assembly is required for Bcl10 nucleation. Given their inability to nucleate and their uniform size, we suspect the observed CARD11$^{8-172}$ oligomers represent an "off-pathway", non-helical conformational state, perhaps mediated by domain-swapping as was observed for CARD9$^{2-152}$ (Supplementary Fig. 1C). Utilizing an anti-HA antibody to decorate the Bcl10 filaments, we were also able to identify continuous CARD11$^{8-172}$-Bcl10 filaments (Supplementary Fig. 4G), demonstrating direct templating as the mechanism of enhanced Bcl10 polymerization.

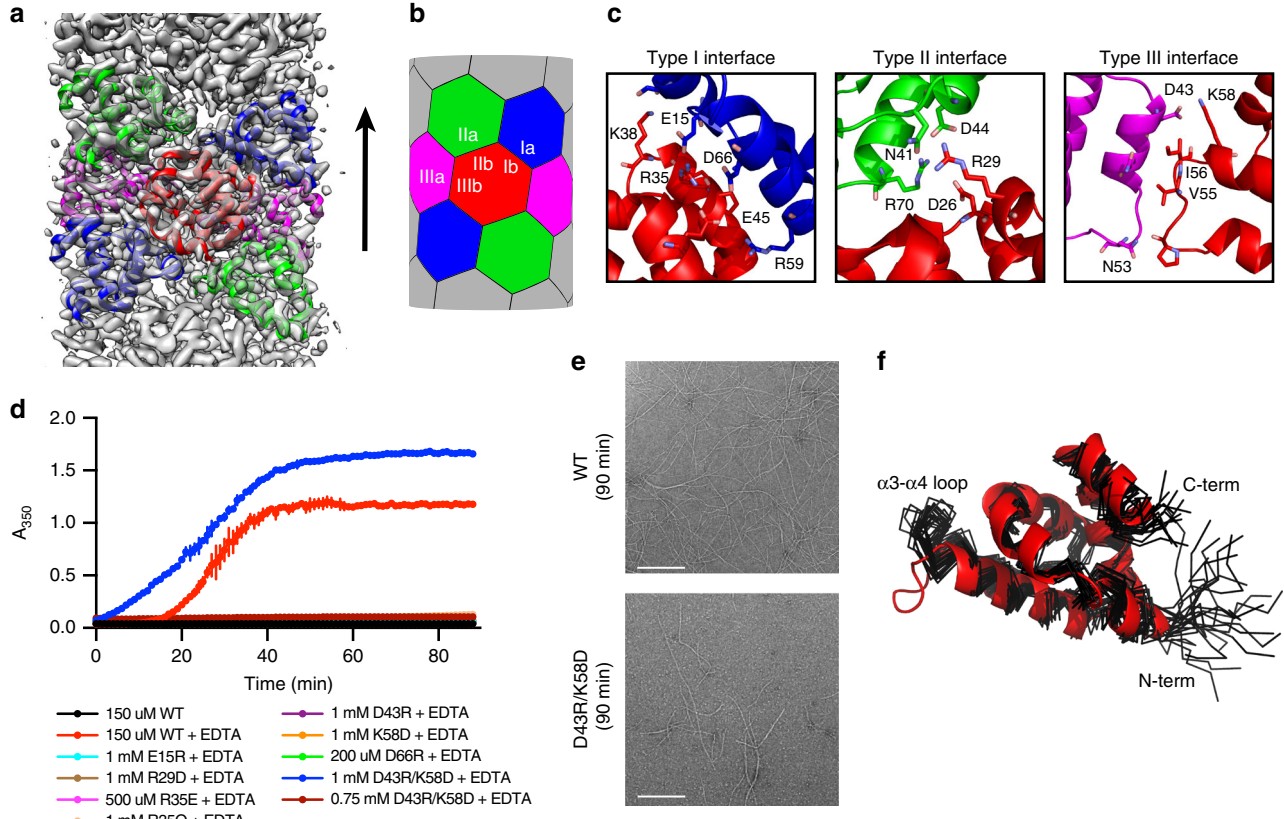

**Fig. 6** The structural basis for unidirectional polymerization of CARD9. **a** Cryo-EM density for CARD9$^{2-152/I107E}$ filaments. Density is only present for the CARD core of the assembly. Seven CARDs of the modeled structure are shown, depicting the six CARDs contacted by any one CARD, colored as in **b**. Arrow indicates the direction of polymerization. **b** Schematic of the type I, II, and III interaction interfaces utilized by the CARD9 CARD. **c** Detail of the CARD interaction type I, II, and III interfaces, with residues involved in the interaction shown as sticks. CARDs are colored as in **b**. **d** CARD9$^{2--97}$ polymerization assay. At time $t = 0$, a super-stoichiometric concentration of EDTA was added to each CARD, which were pre-saturated with 1:1 Zn$^{2+}$. Polymerization was monitored by absorbance at 350 nm. Vertical bars represent the s.d. of three technical replicates. See NS-EM endpoint images in **e** and Supplementary Fig. 6A. For mutants D66R and R35E, Zn$^{2+}$ removal at higher concentrations does lead to increased A$_{350}$, but without visible filaments by NS-EM (Supplementary Fig. 6B), indicating that the mutations are destabilizing and lead to non-specific aggregation upon Zn$^{2+}$ removal at the higher concentrations. Source data are provided in Source Data file. **e** NS-EM micrographs of CARD9$^{2-97/WT}$ and CARD9$^{2-97/D43R/K58D}$ generated at the endpoint of the assay in **d**. Scale bars are 500 nm. **f** Alignment of the apo CARD9 CARD NMR structure (black, PDBID 6E26 [http://dx.doi.org/10.2210/pdb6E26/pdb]) and CARD Cryo-EM structure (red)

### The structural basis for polymerization of CARD9.

As previously reported[31], CARD9$^{2-97}$ filaments present almost exclusively as large bundles when frozen for Cryo-EM. With the CARD9$^{2-152/I107E}$ construct, however, we found that the filaments now remained predominantly singular when frozen shortly after inducing filament formation. With this sample, we were able to generate a 4.0-Å reconstruction of the CARD9 CARD filament (Fig. 6a and Supplementary Fig. 5). The coiled-coil domain is disordered relative to the CARD core, such that the reconstruction contains only CARD residues 8–95. The overall resolution was estimated at 4.0-Å by Fourier shell correlation (FSC, Supplementary Fig. 5B), with regions towards the center of each CARD estimated at ~3.5 Å and the N-terminal region and flexible α3-α4 loop estimated at ~4.5 Å (Table 2 and Supplementary Fig. 5C). The map was of sufficient quality to resolve most sidechains (Supplementary Fig. 5E), allowing us to model the CARD and define the interfaces involved in the assembly. The CARD adopts the canonical helical architecture of a death domain, with helical symmetry of a 5.11 Å rise and 101.6° left-handed rotation, consistent with a growing body of literature demonstrating a narrow range of parameters adopted by helical assemblies of CARDs (4.85–5.13 Å rise and 100.2°−101.6° left-

handed rotation) including that of Bcl10[36,38–41]. As is typical of this architecture, the CARD utilizes three CARD-CARD interfaces, known as the Type I, Type II, and Type III interfaces, corresponding to the 3-, 4-, and 1-start helical symmetries, respectively (Fig. 6a, b). As detailed in Fig. 6c, interactions within the interfaces are predominantly electrostatic in nature, consistent with our previous observation that increasing salt concentration can inhibit assembly[31].

To confirm the importance of the identified interfaces in CARD helical assembly, we generated a series of mutants in each interface and tested their impact on the CARD filament formation. As we have previously demonstrated, upon removal of Zn$^{2+}$ from 150 μM 1:1 Zn$^{2+}$-bound CARD9$^{2-97/WT}$ by addition of EDTA, the CARD assembles into filaments within ~20 min (Fig. 6d, e). As monitored by A$_{350}$ and NS-EM, mutation of residues in the Type I (E15R, R35E, D66R), Type II (R29D), or Type III (D43R, K58D) interfaces completely blocked filament formation when tested up to a 1 mM concentration (Fig. 6d and Supplementary Fig. 6A, B). We also tested a clinically identified mutant (R35Q), which was found in a patient with invasive fungal infections in the central nervous system and digestive tract[18]. As with the other interface mutants, R35Q completely

**Table 2 Structural statistics for the CARD9 CARD helical assembly**

| | |
|---|---|
| Magnification | 130,000 |
| Voltage (kV) | 300 |
| Electron exposure (e–/Å$^2$) | 50.9 |
| Defocus range (μm) | 1.25–2.5 |
| Pixel size (Å) | 1.085 |
| Symmetry imposed | Helical: 5.11 Å, −101.6° |
| Initial particle images (no.) | 141,592 |
| Final particle images (no.) | 31,908 |
| Map resolution (Å) | 4.0 |
| FSC threshold | 0.143 |
| Map resolution range (Å) | 3.5–4.5 |
| Initial model used (PDB code) | 6E26 |
| Model resolution (Å) | 4.0 |
| FSC threshold | 0.143 |
| Map sharpening B factor (Å$^2$) | −90 |
| Model composition | |
| Non-hydrogen atoms | 7190 |
| Protein residues | 880 |
| Mean B factor (Å$^2$) | 64.8 |
| R.m.s. deviations | |
| Bond lengths (Å) | 0.007 |
| Bond angles (°) | 0.789 |
| Validation | |
| MolProbity score | 2.03 |
| Clashscore | 7.4 |
| Poor rotamers (%) | 0.0 |
| Ramachandran plot | |
| Favored (%) | 87.2 |
| Allowed (%) | 13.8 |
| Disallowed (%) | 0.0 |

blocked CARD filament formation, providing a molecular explanation for the clinically observed deficiency in CARD9 signaling. Within the Type III interface, D43 and K58 form a relatively isolated salt bridge. By swapping the charges on these two residues (CARD9$^{2−97/D43R/K58D}$), the CARD regains filament-forming capacity, albeit at a higher protein concentration, confirming the importance of this salt-bridging interaction in the formation of the CARD assembly.

Comparing the Cryo-EM structure of CARD9$^{2−152/I107E}$ to a previously determined NMR structure of CARD9$^{2−97}$ (PDB ID 6E26), most of the CARD remains minimally changed between the monomeric and polymerized form (Fig. 6f). The exception is the flexible α3-α4 loop, which adopts a 'closed' conformation in the NMR structure, but an "open" conformation in the filament, stabilized by interactions across the Type III interface. The Zn$^{2+}$-binding site, comprising critical residues Cys10 and His73, is also minimally perturbed upon filament formation (Supplementary Fig. 6C), failing to resolve the outstanding question as to how Zn$^{2+}$ binding modulates CARD9-CARD filament assembly.

Given the structural similarity of the monomeric CARD9 and CARD11 CARDs[31], we anticipated that the CARD11 CARD (CARD11$^{8−109}$) forms a helical assembly in a similar manner. To probe the CARD11 CARD helical assembly, we used Rosetta to generate a model of the CARD11$^{8−109}$ filament based on our EM structure of the CARD9 CARD filament. As shown in Supplementary Fig. 7A, our model predicts utilization of the same three canonical interfaces; while the residues involved in the interactions are not strictly conserved with those of CARD9, the interaction remains predominantly electrostatic. To experimentally validate our model, we generated mutations within each of the interfaces at residues involved in putative CARD-CARD

interactions. Unlike the WT CARD11 CARD, which forms filaments in vitro at 250 μM, each of the mutants were incapable of filament formation at double that concentration (Supplementary Fig. 7B), confirming their relevance in mediating the CARD11 CARD helical assembly.

A number of CARD domains, including those of Caspase-1 and Bcl10, have been demonstrated to polymerize unidirectionally[36,39], implying that only one end of a growing filament is structurally compatible with recruiting additional monomers. To test whether CARD9 CARDs likewise exhibit unidirectional growth, we induced CARD9$^{2−97/WT}$ filament formation in the presence of an equimolar concentration of interface mutants, demonstrated in Fig. 6d to be unable to form filaments. If the "b" interface tip (top as oriented in Fig. 6a) of the filament can recruit additional monomers, mutants in the "a" interfaces would be unable to effectively bind growing CARD9$^{2−97/WT}$ filaments, while mutants in the "b" interface would be able to bind and "cap" growing CARD9$^{2−97/WT}$ filaments, inhibiting further growth (See Supplementary Fig. 6D for schematic cartoon). We found that the three "a"-interface mutants tested minimally impacted CARD9$^{2−97/WT}$ filament growth, while the three "b"-interface mutants significantly impaired CARD9$^{2−97/WT}$ filament growth, demonstrating that the CARD9 CARD polymerizes unidirectionally, with the 'b'' interface at the leading end. The directionality of this growth is consistent with the directionality previously suggested for Bcl10.

## Discussion

The existence of an autoinhibited state of CARD11 was first indicated by Sommer et al.[8], who demonstrated that phosphorylation of the ID relieves apparent autoinhibition of the protein. The biological and pathological importance of autoinhibition was reinforced in a series of studies demonstrating that oncogenic, constitutively activating mutations functionally disrupt this repression[30,32,42]. However, the specific mechanism by which CARD11 and the rest of the protein family are held in an autoinhibited state prior to activation has remained an open question.

Here, we have determined the structure of an autoinhibited CARD9 construct, demonstrating that the CARD forms an extensive, predominantly hydrophobic, interaction with the adjacent linker, and coiled-coil elements (Fig. 1). The residues involved in this interaction are largely conserved, both between CARD9 and CARD11 and among the four-member family that also includes CARD10 (CARMA3) and CARD14 (CARMA2) (Supplementary Fig. 1G), suggesting that the nature of the autoinhibited state is conserved throughout the family. Further in support of this notion, nearly all of the activating mutations previously identified in CARD11 map to this interface, and in particular to "inward-facing" residues; for example, in the linker, the only residues not identified as activating are the solution-exposed residues in the short helical segment (Fig. 2b).

With insight provided by this structural understanding of the autoinhibited state, we demonstrated that the CARD11 interface is more labile than that of CARD9, but is reinforced by the ID. Replacing the CARD and CARD-interacting region of CARD11 with the more stable region of CARD9 (Fig. 3b) revealed that the ID is functionally redundant in maintaining autoinhibition in the context of a robust CARD-coiled-coil interaction. Jattani et al.[9] recently demonstrated that the CARD11 ID comprises numerous redundant sections that interact in a multivalent manner with the CARD and the larger coiled-coil domain, explaining why activating mutations within the ID itself have not been identified[9,43]. Following similar logic, we anticipate that the multivalent nature of the ID interaction precludes single mutations in the CARD or coiled-coil domain from functionally disrupting all interactions

with the ID; yet, disruptions to the CARD-coiled-coil interface that promote helical assembly of the CARD may block interactions with the ID, thus explaining the reduced ID-CARD co-immunoprecipitation in the presence of CARD-coiled-coil-disrupting mutants previously reported. Further investigation of the CARD11 ID, CARD, and coiled-coil will be required to detail the specific structural determinants of this complex, multivalent interaction and how upstream signaling events disrupt this repressed state to fully activate CARD11.

For CARD9, activation is mediated by ubiquitination at Lys125, which displaces the CARD from the coiled-coil to a comparable extent as an activating mutation (Fig. 4). Cao et al.[12] showed that the ubiquitin ligase TRIM62 activates CARD9 via a specific ubiquitination at Lys125 and that, upon TRIM62 over-expression, CARD9 is heavily modified via K27-linked ubiquitin, but did not investigate whether poly-ubiquitination is required for CARD9 activation. Our studies (Fig. 4 and Supplementary Fig. 2) suggest that conjugation of a single ubiquitin to Lys125 is sufficient to disrupt the CARD-coiled-coil interface in a similar manner and to a similar degree as mutations that are activating in cells. We anticipate that poly-ubiquitination could function to promote more robust signaling through CARD9 by providing additional hydrophobic ubiquitin surfaces to compete with the CARD for coiled-coil binding and/or slowing inactivation by deubiquitinases.

Another activating post translational modification described for CARD9, phosphorylation at Thr231, occurs outside the region of the protein structurally characterized here, further into the larger coil-coil domain. Given that disruption of the CARD-coiled-coil interface is sufficient alone to activate CARD9 upon overexpression (Fig. 2), we suggest that the T231 site may play a role in modulating coiled-coil interactions. In particular, we note that the polyvalent nature of the multiple coiled-coil-forming segments in CARD9 (Fig. 1a) may allow the protein to partition into phase-separated condensates[44] that would be expected to promote the formation of nucleating seeds by increasing the local concentration of the CARD. In support of this theory, Qiao et al.[15] found that a CARD11 construct comprising this region (CARD11$^{8-302}$) purifies as a large multimer capable of nucleating Bcl10 polymerization. Furthermore, a second "hot-spot" of CARD11 oncogenic mutations exists near this site[25], including the homologous residue, Ser250, suggesting it as a conserved regulatory region of the protein.

Our findings here largely support and expand upon the pivotal studies by Qiao et al.[15], which first demonstrated CARD11 seeding of Bcl10 polymerization. We do, however, note a discrepancy in our findings. Namely, Qiao et al.[15] reported that the CARD11$^{8-172}$ construct did not require higher-order multi-merization to nucleate Bcl10 polymerization, while our observations for both CARD9$^{2-152}$ and CARD11$^{8-172}$ suggest that the proteins are capable of seeding Bcl10 nucleation only after assembling into a helical template (Fig. 5 and Supplementary Fig. 4). In our studies, the CARD11$^{8-172}$ dimer is capable of forming Bcl10-nucleating filaments upon concentration and incubation, suggesting that Qiao et al.[15] may have handled the CARD11$^{8-172}$ protein in such a way as to inadvertently allow some nucleating multimers to form subsequent to SEC purification.

The structures of the CARD9$^{1-142}$ dimer and the CARD9 CARD helical assembly we present here provide insight into both the autoinhibited and activated forms of this family of Bcl10-nucleating proteins. In addition to providing a structural explanation for pathogenic mutations in both CARD9 and CARD11, these structures allowed us to define the mechanisms of activation for both proteins in the course of canonical signaling. While significant questions remain, in particular in understanding the

regulatory role of the larger coiled-coil domain, our study provides a number of critical steps towards a full structural description of regulation within the protein family.

## Methods

**Protein purification.** All purified CARD9, CARD11, and ubiquitin constructs comprise the human sequences and were expressed in BL21(DE3) cells with an N-terminal TEV-cleavable 6xHis tag. Protein production was achieved by growth for 48–72 h at 16 °C in Terrific Broth autoinduction media or $^{15}$N autoinduction media for unlabeled and $^{15}$N-labeled protein, respectively[45]. Uniform $^{13}$C$^{15}$N-labeled and $^{2}$H$^{13}$C$^{15}$N-labeled protein was generated by induction in $^{13}$C$^{15}$N or $^{2}$H$^{13}$C$^{15}$N minimal media with 0.5 mM isopropyl β-D-1-thiogalactopyranoside (IPTG) for 6 h at 37 °C.

Ub$^{G76C}$ was purified by lysing cell pellets in nickel buffer (50 mM HEPES, 500 mM NaCl, 20 mM imidazole, pH 7.5) via sonication. After a high-speed spin, the soluble fraction was passed over a Ni-NTA (Quiagen) affinity column, washed with nickel buffer, and eluted with nickel buffer supplemented with 400 mM imidazole. TEV protease was added to this solution, which was then dialyzed overnight to into nickel buffer at 30 °C to cleave the 6xHis tag. A majority of the uncleaved Ub$^{G76C}$ was removed by an additional Ni-NTA column, however a small amount of uncleaved Ub$^{G76C}$ nonetheless remained (see Supplementary Fig. 2A). Ub$^{G76C}$ was finally purified over a Superdex 75 gel filtration column (GE Healthcare).

A significant percentage of CARD11$^{2-172}$, CARD9$^{2-152}$ (WT and mutant constructs), and CARD9$^{2-142}$ remained in the insoluble fraction upon gentle lysis; these constructs were thus purified under denaturing conditions. Cell pellets were lysed in 5 M guanidine hydrochloride, 100 mM HEPES, 100 mM NaCl, pH 7.5. After a high-speed spin, the soluble fraction was passed over a Ni-NTA affinity column, washed with urea wash buffer (6.5 M Urea, 50 mM HEPES, 50 mM NaCl, 20 mm imidazole, pH 7.5), and eluted with urea wash buffer supplemented with 400 mM imidazole. Five millimolar TCEP was added to eluate, which was then dialyzed twice against nickel buffer with 0.25 mM TCEP to allow for refolding, followed by addition of TEV protease overnight to remove the 6xHis tag. Uncleaved protein and TEV protease were subsequently removed by Ni-NTA. Finally, all proteins were purified via a Superdex 200 gel filtration column (GE Healthcare). For formulations containing zinc, a saturating concentration of ZnCl$_2$ was added prior to the final gel filtration column.

WT and most mutants of CARD9$^{2-97}$ and CARD11$^{8-109}$ were purified as Ub$^{G76C}$, except that TEV protease cleavage was achieved through overnight incubation at 4 °C and saturating ZnCl$_2$ concentrations were added to CARD9$^{2-97}$ prior to the final Superdex 75 gel filtration column. CARD9$^{2-97}$ mutants that were predominantly insoluble upon gentle lysis were purified under denaturing conditions, as described for CARD9$^{2-152}$ except that the final gel filtration column was a Superdex 75.

The MBP-Bcl10 construct comprises an N-terminal MBP, followed by a TEV cleavage site, human Bcl10, with mutations C10A and C29A to prevent fluorophore labeling of the Bcl10 CARD, and a C-terminal HA-tag (GSGSYPYDVPDYA). The protein was expressed in BL21(DE3) cells grown in LB media at 37 °C to an optical density at 600 nm of 0.7. 0.2 mM IPTG was added to induce protein production for one hour at 37 °C. MBP-Bcl10 pellets were lysed in nickel buffer via sonication and spun down. The soluble fraction was bound to Ni-NTA (Quiagen) affinity column, washed with nickel buffer, and eluted in 3 ml nickel buffer supplemented with 400 mM imidazole. Two micromolar Alexa Fluor 488 C$_5$ maleimide (ThermoFisher) was added to the eluate, which was incubated for 10 min at room temperature. The eluate was then purified over a Superdex 200 gel filtration column. The monomeric MBP-Bcl10 peak was collected, stored at 4 °C, and utilized for FP assays or microscopy within 2 h of elution from the gel filtration column with no concentration during any step of the purification.

For all purifications, the subsequent downstream application dictated the final gel filtration buffers, which are described for each application in the following methods sections.

**NMR data collection.** All NMR spectra shown were collected in 50 mM HEPES, 300 mM NaCl, 0.5 mM TCEP, pH 7.0 at 37 °C on a Bruker 800 MHz Bruker Spectrometer with a cryogenically cooled probe. All were $^{15}$N-TROSY experiments, with the exception of the CARD9$^{2-97}$ spectrum in Fig. 1c, which was a $^{15}$N-HQSC. The CARD11$^{8-172}$ spectrum in Fig. 1b was collected at 150 μM, while the CARD9$^{2-152}$ and CARD9$^{2-97}$ spectra depicted in Fig. 1c were collected at 200 and 400 μM, respectively. The spectra depicted in Fig. 4 and Supplementary Fig. 2 were all collected at 200 μM using identical experimental parameters.

**NMR assignments and solution structure determination.** CARD9$^{2-142}$ assignments and structure determination utilized a construct comprising residues 2–142 of human CARD9. Assignments were performed on 1 mM samples with 1:1 ZnCl$_2$ in 50 mM HEPES, 300 mM NaCl, 0.5 mM TCEP, pH 7.0, 10% D$_2$O, at 37 °C, with data collected on a 600 or 800 MHz Bruker Spectrometer with a cryogenically cooled probe. Backbone assignments were determined through sequential assignment of $^{2}$H$^{13}$C$^{15}$N CARD9$^{2-142}$ using $^{15}$N-TROSY, (HNCO)CACB(CO)NH, HNCACB, NHCA, HNCO, and HN(CA)CO experiments, all utilizing TROSY.

Side chain assignments were determined with $^{13}C^{15}N$ CARD9$^{2−142}$ using $^{13}C$-HSQC (aliphatic and aromatic), (H)CC(CO)NH, HCCH-TOCSY, HCCH-COSY, and $^{13}C$-NOESY-HSQC (aliphatic and aromatic, 120 ms mixing times) experiments, all collected in 99.9% $D_2O$ buffer, as well as a $^{15}N$-NOESY-TROSY experiment collected in 10% $D_2O$. Stereospecific assignments of valine and leucine methyl groups were determined by generating protein with a 1:9 $^{13}C$/unlabeled glucose ratio and collection of $^{13}C$-HSQC spectra as described by Senn et al.[46].

To assist in structure determination, intermolecular distance restraints were collected using an intermolecular $^{13}C$-NOESY-HSQC experiment that allows for selective detection of NOEs between $^{13}C$-and $^{12}C$-bound protons[47]. The sample for this experiment was prepared by mixing equimolar concentrations of denatured $^{13}C^{15}N$ and unlabeled CARD9$^{2−142}$ after elution from the Ni-NTA column and continuing with an otherwise identical purification protocol. The sample was concentrated to 2 mM for data collection. As only 25% of the sample as prepared is detectable by this experiment (i.e., only the $^{13}C^{15}N$ protein in a labeled/unlabeled dimer), and ~1% of the carbons in the sample are differentially labeled (i.e., $^{13}C$ in an "unlabeled" protein due to natural abundance or $^{12}C$ in a "labeled" protein due to a lack of complete isotopic purity in $^{13}C$-glucose), intermolecular NOEs were included conservatively, only when peaks were relatively intense as compared to a standard $^{13}C$-NOESY-HSQC collected on a fully labeled sample. Concentration to 2 mM led to substantial line-broadening due to protein–protein interaction; this experiment was therefore collected with 900 mM NaCl rather than 300 mM to mitigate these interactions. Only cross-peaks for which minimal chemical shift perturbations were observed between 300 and 900 mM NaCl were included in the structure calculation.

N–H residual dipolar coupling (RDC) restraints were also collected on the CARD9$^{2−142}$ dimer. Samples were generated with 800 mM $^2H^{13}C^{15}N$ CARD9$^{2−142}$, with or without 14 mg/ml filamentous Pf1 bacteriophage (Asla Biotech). TROSY and anti-TROSY peaks were collected in an IPAP manner. RDC values were determined as the difference in $J_{NH}$ between the sample with and without added bacteriophage. The alignment tensor was determined iteratively during structure determination using the CYANA v3.97 FitTensor protocol[48].

An additional aliphatic $^{13}C$-NOESY-HSQC (120 ms mixing times, dissolved in 90% $H_2O$/10% $D_2O$ buffer) experiment was collected to assist in structure determination. All spectra were referenced directly (proton) or indirectly (nitrogen and carbon) to an internal DSS standard. All spectra were processed using Bruker TopSpin v3.5 and subsequently analyzed in CcpNMR Analysis v2.4[49].

For structure determination, dihedral angles were estimated using TALOS + [50]. Restraints were enforced to maintain coordination of zinc ions by Cys10 Sγ and His73 Nδ1 and to maintain the zinc ion in-plane with the histidine aromatic ring for each monomer in the structure. Symmetry restraints were included for residues 6–142. NOE peaks were assigned and initial structure determination was achieved using the CYANA v3.97 NOE assignment and structure determination package[48,51]. Sum of $r^{−6}$ averaging was used for all NOEs. For each round of CYANA NOE assignment and structure determination, 100 structures were generated, with the 20 lowest target function structures proceeding to the next round. After the final round of NOE assignments, 100 structures were calculated and subsequently refined in explicit water using the PARAM19 force field in CNS v1.2[52,53] and the WaterRefCNS package developed by Robert Tejero. The 20 lowest energy structures after refinement in water are presented here. Structures were evaluated using PROCHECK-NMR, with statistics presented in Table 1. Surface area calculations were performed using PDBePISA (www.ebi.ac.uk/pdbe/pisa).

Backbone amide assignments were transferred to CARD9$^{2−152/SSC}$ from CARD9$^{2−142}$ with the help of HNCA, HNCB, (HNCO)CACB(CO)NH, and HN(CA)CO experiments collected on $^2H^{13}C^{15}N$-labeled CARD9$^{2−152/SSC}$ in 50 mM HEPES, 300 mM NaCl, 0.5 mM TCEP, pH 7.0. We found CARD9$^{2−152/SSC}$ to be somewhat less stable then CARD9$^{2−152}$ or CARD9$^{2−142}$ during extended data collection; as such, assignment spectra for CARD9$^{2−152/SSC}$ were collected at 0.5 mM at 25 °C. Assignments were then transferred to the 37 °C spectrum by collection of a series of $^{15}N$-TROSY experiments between 25 °C and 37 °C.

Ub$^{G76C}$ backbone amide chemical shifts were assigned with the help of previously determined WT ubiquitin assignments (BMRB 6457[54]) along with a $^{15}N$-NOESY-HSQC collected at 37 °C on 0.7 mM Ub$^{G76C}$ in 50 mM HEPES, 300 mM NaCl, 0.5 mM TCEP, pH 7.0.

All structural depictions were generated in Pymol (pymol.org).

**Mammalian expression constructs.** CARD9 and CARD11 constructs were all generated in the pCMV6-XL4 mammalian expression vector (See Supplementary Table 1 for primers used in this study). CARD9 constructs comprise the full human sequence (amino acids 1–536). For CARD11, we generated the identical sequence as used by McCully and Pomerantz[32], comprising the mouse CARD11 sequence 1–1159 (see Supplementary Fig. 1B for alignment of the mouse and human sequences for residues 1–172), with residues 441–671 removed in the ΔID construct. Chimeric constructs CARD11$^{1−143}$/CARD9 and CARD9$^{1−131}$/CARD11-ΔID were generated by interchanging residues 1–131 of CARD9 and 1–143 of CARD11.

**Mammalian cell culture and NF-κB reporter assay.** HEK-Blue-hNOD2 reporter cells (HEK293 cells expressing an optimized secreted embryonic alkaline phosphatase (SEAP) reporter gene, InvivoGen hkb-hnod2) were cultured in 50:50 F12

DMEM medium supplemented with 10% FBS, penicillin, streptomycin and 2 mM glutamine at 37 °C with 5% $CO_2$. Cells were routinely screened for mycoplasma contamination. HEK-Blue-hNOD2 reporter cells were used to quantitatively measure NF-κB activation. CARD9, CARD11, and chimeric constructs were transiently transfected with TransIT-LT1 (Mirus, Madison, WI, USA) for 48 h. Activation of NF-κB in reporter cells leads to production of SEAP that was quantified using HEK-Blue$^{TM}$ Detection (InvivoGen), a cell culture medium that allows for real-time detection of SEAP.

Lysates were generated via lysis in RIPA buffer. Total protein content was measured via BCA, samples were boiled in LDS with reducing agent, and 25 µg protein per sample was loaded run on an sodium dodecyl sulfate polyacrylamide gel electrophoresis gel followed by transfer to a nitrocellulose membrane. Samples were probed via western blot analysis using a CARD9 polyclonal antibody that targets amino acids 521 to 536 of human CARD9 (1:1000 dilution, ThermoFisher PA5–19993), a CARD11 polyclonal antibody that targets residues surrounding residue 362 (1:1000, dilution, Cell Signaling 4440S), or a monoclonal β-actin antibody, (1:1000 dilution, Licor 926–42212). Blots were visualized using an HRP-conjugated mouse (1:5000 dilution, Sigma NA931V, for β-actin) or rabbit (1:5000 dilution, Sigma NA934V, for CARD9 and CARD11), SuperSignal West Pico Chemiluminescent Substrate (ThermoScientific), and a PXi6 imager (Syngene) or film.

**Ubiquitin conjugation.** CARD9$^{2−152/SSC}$ and Ub$^{G76C}$, each purified in 50 mM HEPES, 150 mM NaCl, 1 mM TCEP, pH 7.0, were mixed at a 1:8 to 1:10 ratio to bias the conjugation against CARD9$^{2−152/SSC}$—CARD9$^{2−152/SSC}$ conjugation. This mix was passed over a HiTrap desalting column (GE Healthcare) equilibrated in 50 mM HEPES, 150 mM NaCl, pH 7.0. BMOE was solubilized at 20 mM in DMSO and added incrementally to the solution, with 0.1 molar equivalents (as compared to the total protein concentration) added at a time, followed by mixing and 30 s of incubation at room temperature. BMOE was added in this manner past saturation, to a final concentration of 0.7 molar equivalents, followed by neutralization by excess β-mercaptoethanol. The desired CARD9$^{2−152/SSC}$—Ub$^{G76C}$ conjugate was purified from other off-target products with a Superdex 200 gel filtration column (GE Healthcare) in 50 mM HEPES, 300 mM NaCl, 0.5 mM TCEP, pH 7.0 (See Supplementary Fig. 2A).

**CARD9$^{2−152}$ and CARD11$^{8−172}$ filament formation.** CARD9$^{2−152}$ constructs were purified bound 1:1 to $Zn^{2+}$ in 50 mM HEPES, 150 mM NaCl, 0.5 mM TCEP, pH 7.0. A twofold excess of EDTA was added to the indicated concentrations of protein, followed by incubation at the indicated temperature for the indicated time. CARD11$^{8−172}$ was purified in 20 mM Tris, 150 mM NaCl, 0.5 mM TCEP, pH 7.5, followed by incubation at either 4 µM or 1 mM as indicated for 16 h at 37 °C.

**Bcl10 fluorescence polarization assay.** The assay was conducted as previously described[31] in 20 mM Tris, 150 mM NaCl, 0.5 mM TCEP, pH 7.5 in a 20 µl final volume. Purified MBP-Bcl10 sparsely labeled with AF488 $C_5$ malemide (see above) was incubated alone or with addition of CARD9 or CARD11 constructs prepared as indicated in the text. For indicated samples, TEV protease was added to 0.05 mg/ml at the initiation of the assay. Bcl10 polymerization was monitored by measuring fluorescence polarization at 519 nm while exciting at 495 nm on a Molecular Devices SpectraMax M5e plate reader at 25 °C.

**NS-EM sample preparation and imaging.** All NS-EM samples were incubated on glow-discharged carbon on 400-mesh copper grids (Electron Microscopy Sciences), followed by staining with 2% uranyl acetate. Images were collected on a Talos F200C microscope operated at 200 kV equipped with a Ceta camera (Thermo-Fisher) at 2.006 Å/pixel (Supplementary Fig. 4D, right), 4.097 Å/pixel (Fig. 5 and Supplementary Figs. 3, 4A, 4D left/center, 4E, and 7), or 10.6 Å/pixel (Fig. 6 and Supplementary Fig. 6). CARD9$^{2−152}$ samples in Fig. 4a were diluted to 50 µM just before grid preparation. CARD11$^{8−172}$ samples in Supplementary Fig. 4A were diluted to 4 µM just before grid preparation. CARD11$^{8−109}$ samples depicted in Supplementary Fig. 7 were diluted to 100 µM just before grid preparation. Grids in Supplementary Fig. 4D, E were prepared directly from SEC fractions (Supplementary Fig. 4C, green) without dilution. Samples in Fig. 5f–h and Supplementary Fig. 4G were prepared identically as in the corresponding FP assay, with 2 µM MBP-Bcl10, 2 µM CARD9$^{2−152/I107E}$ filaments, 10 µM CARD9$^{2−152/L115I}$ filaments, and/or 2 µM CARD11$^{8−172}$ filaments as indicated and with addition of 0.05 mg/ml TEV protease. After a 2-min incubation at room temperature, 4 µl of sample was transferred to the EM grid. For samples including antibody, monoclonal anti-HA antibody (Sigma H3663) was added to a concentration of 0.11 mg/ml on the grid, followed by a 1-min incubation before blotting and staining. CARD9$^{2−97}$ samples were prepared without dilution at the 90-min endpoint of the polymerization assays depicted in Fig. 6d and Supplementary Fig. 6B.

**Hydrogen-deuterium exchange.** Samples, purified in 50 mM HEPES, 300 mM NaCl, 0.5 mM TCEP, pH 7.0, were diluted ~15-fold into 10 mM phosphate buffered deuterium oxide solution of pH 7.0 with 150 mM NaCl and allowed to exchange at 20 °C for times ranging from 30 s to 1 day. Exchange was arrested by 1:1 dilution with a pH 2.3 buffer containing 400 mM Glycine and 4 M

Guanidinium chloride. For measurement of carried deuterium, 100 μl of this mixture was immediately passed through an immobilized pepsin column (2.1 × 30 mm, Applied Biosystem), peptides produced were bound to an online trap column for desalting (Acquity Vanguard C8), separated by a reverse-phase chromatography (Acquity UPLC BEH C18, 1.7 μm particle size, 1.0 × 50 mm) all at 0 °C, and then introduced into the mass spectrometer by electrospray ionization (Thermo-Fisher Orbitrap XL, 60k resolution at $m/z$ 200). A LEAP Pal XT robotics platform was used to automate labeling, quenching, and liquid handling. Measurements generally followed protocols described previously[55,56]. Extracted ion chromatograms were produced by the ExMS program[57] and analyzed by custom python scripts[58] to extract the amount of carried deuterium by each peptide at each time, produce uptake plots, and determine protection factors using an empirical method described previously[59]. See Source Data for raw uptake measurements, protection factors, and protection factor ranges for all peptides.

**Cryo-EM data collection and analysis**. In all, 0.5 mM CARD9$^{2-152/I107E}$, purified in 50 mM HEPES, 150 mM NaCl, 0.5 mM TCEP, pH 7.0 bound 1:1 to Zn$^{2+}$, was incubated for 10 min at 25 °C after addition of 1 mM EDTA. This solution was diluted to 0.1 mM in the same buffer, 3.5 μl of which was added to glow-discharged Protochips C-Flat 2/1 200 mesh holey carbon grids. After a 45 s incubation, the grid was washed and blotted six times with buffer, followed by a final 3.5 μl buffer addition and plunge freezing in liquid ethane using a Vitrobot Mark IV (ThermoFisher).

The sample was imaged on a Titan Krios Cryo Transmission Electron Microscope (ThermoFisher) operated at 300 kV and equipped with a K2 Summit direct electron detector camera, and a Bioquantum energy filter (Gatan). Movies were recorded at a nominal magnification of 130,000, corresponding to 1.085 Å/pixel. Forty frames were collected over 10 s, with an exposure rate of 5.093 e/Å$^2$/s at between 1.25 and 2.5 μm defocus with an energy slit width of 20 eV.

See Supplementary Fig. 5A for a graphical depiction of the refinement scheme. Four-thousand four-hundred and three movies were collected and aligned using cisTEM[60]. Twelve-thousand three-hundred and eighty-six individual filaments were picked by hand with the EMAN2 program e2helixboxer[61], the positions of which were imported into RELION 2.1[62], and extracted into 141,592 particles using a 30 Å shift between particles. Particle positions were imported into cisTEM. The contrast transfer function (CTF) for each movie was determined in cisTEM and only those with good CTF fits to at least 5 Å were included for further analysis, yielding 802 images with picked filaments. Three rounds of two-dimensional classification were carried out in cisTEM; only classes with layer lines visible to better than ~6 Å were carried through to the following round, yielding 31,908 particles. These particles were used to generate an ab initio 3D model of the filament, which was imported into RELION along with the particle stack. Helical refinement was carried out in RELION[63], beginning with the helical parameters previously determined for a NS-EM reconstruction of the CARD9 CARD filament[31] using the Spring[64] routine Segclassreconstruct, 5 Å rise, 102° rotation. The helical parameters were optimized in RELION by searching between a 4.8 to 5.4 Å rise and a −100° to −104° rotation, yielding final parameters of a 5.11-Å rise and −101.6° rotation, along with a ~6.0 Å reconstruction. This volume was imported back into cisTEM, in which further refinement in C1 (no helical symmetry applied) was carried out, leading to a ~4.2 Å reconstruction. This volume and particle stack were finally exported into helical Frealign 9.11[65,66], using the refined helical parameters determined in RELION. Using a 5.0-Å high-resolution limit for particle alignment, refinement in Frealign yielded a 4.0-Å reconstruction (see Supplementary Fig. 5B for FSC curve). A local resolution estimate was determined using local_resolution, a re-implementation (Rohou, manuscript in preparation) of blocres[67]. The reconstruction was sharpened in cisTEM using a pre-cutoff B-factor of −90 Å$^2$.

"Fit in map" in UCSF Chimera[68] was used to fit the previously determined, lowest energy NMR solution structure (PDB ID 6E26) into the sharpened density and to generate nine symmetry-mates according to the determined helical parameters. The map was iteratively refined in Phenix[69] and COOT[70], maintaining strict non-crystallographic symmetry among the CARDs. Structural depictions were generated in UCSF Chimera or Pymol (pymol.org).

**CARD9$^{2-97}$ filament formation**. The in vitro CARD9$^{2-97}$ polymerization assay (Fig. 6d) was performed in 50 mM Tris, 150 mM NaCl, 0.5 mM TCEP, pH 7.0 as previously described[31]. All CARD9$^{2-97}$ constructs utilized were purified bound to equimolar Zn$^{2+}$ by addition of ZnCl$_2$ prior to the final gel filtration column. Fifty microliters of CARD9$^{2-97/WT}$ or mutants were prepared at the indicated concentrations in a 384-well clear-bottom plate. The assay was initiated by addition of a super-stoichiometric concentration of EDTA as indicated. Polymerization was monitored by measuring optical density at 350 nm on a Molecular Devices SpectraMax M5e plate reader while shaking at 25 °C. At the 90 min endpoint of the assay, NS-EM grids were generated.

**CARD11$^{8-109}$ filament formation**. WT or mutant CARD11$^{8-109}$ constructs were purified in 50 mM HEPES, 50 mM NaCl, 0.5 mM TCEP, pH 7.0, concentrated to either 250 μM or 500 μM, and incubated at 25 °C for 2 h prior to grid preparation.

**DAmFRET plasmids**. CARD9$^{1-98}$, CARD9$^{1-142}$, CARD9$^{1-142/I107E}$ and Bcl10 full-length sequences (Supplementary Table 2) were cloned into V08 as described in Khan et al.[35], for C-terminal tagging with mEos3.1. Protein expression is driven by a GAL1 promoter, with URA3 as selection marker.

**Yeast strains for μNS nucleation assay**. To create artificial intracellular seeds of CARD9 variants and Bcl10, sequences were fused to a constitutive condensate-forming protein, μNS (471–721)[37], hereafter "μNS". Yeast strain rhy1903 was created by replacing the HO locus in rhy1734 with a cassette consisting of: natMX followed by the tetO7 promoter followed by counterselectable URA3 ORFs derived from C. albicans and K. lactis, followed by μNS-mTagBFP2. Yeast strain rhy2068 was constructed identically, except the cassette lacked μNS. To create strains rhy2077, rhy2079, rhy2089, rhy2228, rhy2229, rhy2254, and rhy2255, AseI digests of rhx1140, rhx2138, rhx2303, and rhx2304 were transformed into rhy1903 and rhy2068 to replace the counterselectable URA3 ORFs with the gene of interest. The resulting strains express the proteins of interest fused to mTagBFP2 or μNS-mTagBFP2, under the control of a doxycycline-repressible promoter. Transformants were selected for 5-FOA resistance and validated by PCR. See Supplementary Table 3 for a list of all strains used in this study.

For measuring nucleating interactions strains rhy2077, rhy2079, rhy2089, rhy2228, rhy2229, rhy2254, rhy2255 were maintained on doxycycline (40 mg/ml) until initial culture for DAmFRET assay.

**DAmFRET data collections and analysis**. To evaluate CARD9 variants and Bcl10 nucleation barriers, we applied DAmFRET. This method exploits a photoconvertible fluorophore, heterogeneous expression levels, and large cell numbers to quantify via flow cytometry the frequency of nucleation as a function of protein concentration[35]. Briefly, single transformant colonies were inoculated in 200 μl of SD-URA in a microplate well and incubated in a Heidolph Titramax platform shaker at 30 ºC, 1350 RPM overnight without presence of Dox. Cells were washed with sterile water, resuspended in galactose-containing media, and allowed to continue incubating for approximately 16 h. Microplates were then illuminated for 25 min with 320–500 nm violet light to photoconvert a fraction of mEos3.1 molecules from a green (516 nm) form to a red form (581 nm).

DAmFRET data were collected on a ZE5 cell analyzer cytometer or an ImageStream$^x$ MkII imaging cytometer (Amnis) at x60 magnification. Autofluorescence was detected with 405 nm excitation and 460/22 nm emission; SSC and FSC were detected with 488 nm excitation and 488/10 nm emission. Donor and FRET fluorescence were detected with 488 nm excitation and 425/35 nm or 593/52 nm emission, respectively. Acceptor fluorescence was excited with 561 nm excitation and 589/15 nm emission. Approximately 500,000 events were collected per sample. Data compensation was done in the built-in tool for compensation (Everest software V1.1) on single-color controls: non-photoconverted mEos3.1 and dsRed2 (as a proxy for the red form of mEos3.1).

Data was processed on FCS Express Plus 6.04.0015 software (De Novo). Events were gated for single unbudded cells by FSC vs. SSC, followed by gating of live cells with low autofluorescence and donor positive. Live gate was then selected for double positives (donor and acceptor). Plots represent the distribution of AmFRET (FRET intensity/acceptor intensity) vs. Acceptor intensity (protein expression).

**Sequence alignment**. Multiple sequence alignments were performed using Clustal Omega[71].

**Homology modeling**. The CARD11 sequence was threaded into a monomer of the CARD9 helical filament using Rosetta, followed by generation of a helical assembly by backbone alignment with the CARD9 filament. This assembly was then energy minimized using 20 parallel replicates of the Rosetta relax routine[72]. The lowest energy structure is depicted in Supplementary Fig. 7.

**Reporting summary**. Further information on research design is available in the Nature Research Reporting Summary linked to this article.

## Data availability

Chemical shifts for the CARD9$^{2-142}$ dimer were deposited in the Biological Magnetic Resonance Database (http://www.bmrb.wisc.edu/) under accession code 30543. An electron microscopy density map for the CARD9$^{2-142/I107E}$ filament has been deposited in the Electron Microscopy Data Bank (https://www.ebi.ac.uk/pdbe/emdb/) under accession code EMD-9332. Atomic coordinates of the NMR solution structure of CARD9$^{2-142}$ dimer were deposited in the Protein Data Bank (https://www.rcsb.org/) under accession code 6N2M. Atomic coordinates of a representative array of the CARD9$^{2-152/I107E}$ helical assembly were deposited in the Research Collaboratory for Protein Data Bank under accession code 6N2P. The source data underlying Figs. 2C, 3B, 4B, 4F, 4H, 5C, 6D, and Supplementary Figs. 2F, 3B, 3E, 4B, 4F, 6B, and 6D are provided as a Source Data file.

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

## Acknowledgements

We thank Menno van Lookeren Campagne, Joseph Chavarria-Smith, and Elizabeth Helgason for their helpful discussion and assistance throughout this study. We thank David Hewings for his advice regarding chemical ligation strategies. We thank Claudio Ciferri for his guidance in EM labeling strategies. We thank Christian Cunningham for his guidance in Rosetta homology modeling.

## Author contributions

Experiments were designed by M.J.H., A.W., B.T.W., R.H., A.R., E.C.D., and W.J.F. Research was supervised by R.H., A.R., E.C.D., and W.J.F. M.J.H. collected and analyzed the NMR data and determined the NMR structure. A.W. performed and analyzed data from the NF-κB activation assays. A.R.G. performed and analyzed the DAmFRET experiments. B.T.W. performed and analyzed the hydrogen-deuterium exchange experiments. M.J.H., A.R., and C.P.A. optimized conditions and collected the Cryo-EM data. M.J.H. determined the structure of the CARD9 CARD filament. All other experiments were performed by M.J.H. The manuscript was written by M.J.H. with editing by R.H., A.R., E.C.D., and W.J.F. along with feedback and approval from all authors.
