## [Peer Review File · Nature Communications]

Reviewers' comments:

Reviewer #1 (Remarks to the Author):

In this manuscript, Holliday MJ, et al aims to study the signaling mechanism leads to activation of Bcl10 in both macrophage (by CARD9) and lymphoid cells (by CARD11/CARMA1), using a combination of biochemical and structural methods.

Existing literature:

Both CARD9 and CARD11 belongs to a family of proteins (CARMA1/2/3, CARD9) that share similar protein architecture. In different cell types, these proteins are believed to play similar functions in regulating cellular signaling pathways lead to NF- κ B activation. CARD11/CARMA1 is the most well studied protein in this family, mostly due to its important in governing lymphocyte activation, differentiation and proliferation. In 2013, Qiao Q, et al reported a filamentous structure of Bcl10 at about 7-10Å. In this study, CARMA1, Bcl10 and the downstream MALT-1 were found to form a large signalosome called CMB complex. In 2018, Liron D, et al and Florian S, et al, two manuscripts reported the high resolution (3.5-4Å) structure of Bcl10 filament, and built the atomic model. In 2018, Seeholzer T, et al reported that CARMA1-Bcl10 fusion constructs was signaling active in mammalian cell lines. However, the exact structural features of CARD11 (CARMA1) active signaling complex remains unknown. CARD9 is relatively less studied, mostly known by its function to serve as signaling adaptors downstream of lectin proteins. In 2018, Holliday MJ, et al (same research group of this study) reported a 20Å lower resolution filament model of CARD9. However, atomic model is not built at that time.

There are mainly two findings in this manuscript.

1. The resting state dimer:

This study mainly used CARD9 as the structural representative protein for the entire CARMA family. By using NMR, the authors firstly solved the structure of CARD9 containing an extended structural motif that is not commonly seen in CARD domain structures. The authors then proposed a auto-inhibitory state that is related to the dimer formation of the NMR structure, which further explains why certain post-translation modification could activate CARD9. Then using homologous modeling, the authors proposed that CARD11 was regulated in similar ways. Additional biochemical and cellular assays (mainly several fluorescent based probing of Bcl10 activation assay, and one set of NF- κ B luciferase assay) were used to verify their findings.

'The authors need to address the following concerns in this part. For Figure 5B. Firstly, the author mentioned a sharp increase in polymerization at concentration above 100 μ M, however the inset in figure 5B was determined by vertical dash line at approximately 50 μ M. If the line is to be adjusted to 100 μ M (also why figure S5D is cited here? Maybe referred to figure S4D?). The difference between CARD91-142/107E-mEos3 and CARD91-142/WT-mEos3 does not seem significant when gated at 100 μ M. Additional statistical analysis would be helpful to clarify the statement here. Are the chimeric constructs used in Figure 3 (luciferase assay) behave similarly if using this fluorescent assay or using FP? Comparing the same constructs in different assays would build a strong linkage between biochemical and cellular assays.'

2. CARD9 filament as the activation state

The authors managed to purify CARD9 in its homogenous filamentous state and eventually solved its atomic structure. This is a great achievement in understanding the CARMA family proteins. The structural approaches were appropriately executed, maybe the final density is over-sharpened a bit, but nevertheless a good representation of the density observed. The procedures leading to the identification of the helical symmetry could be better elaborated, perhaps with the facilitation of power spectrum map. Interestingly, the helical symmetry (5Å rise, and -101 degree rotation) matched the Bcl10 helical symmetry perfectly, as well as all the other existing CARD filaments (MAVS-CARD, ASC-CARD, pro-CASP1-CARD, NLRC4-CARD, RIP2-CARD). In addition, this manuscript is also another good example of showing how different upstream CARDS could activate the shared common downstream adaptors, like in the cases of RIG-I/MDA5-MAVS, NOD1/2-RIP2. A insightful addition to the existing literature.

'However, in order to extend their conclusions to CARD9/11-Bcl10, the authors need to demonstrate that these two complexes are indeed similar. This is particularly important to claim CARD11 signaling mechanism in this study, since its structure is still unknown. Maybe they can consider building models of the potential ways how CARD9/11 could interact with Bcl10, or how CARD11 filament should look like (using homologous models), and perform mutagenesis like what they did in CARD9, then using negative stain EM or luciferase assays to test these hypothesis.'

Overall, this is an extensive structural investigation of CARD9 in its presumed resting dimer state and active filamentous signaling state, a similar educated guess was made about CARD11, based on biochemical assays. The comments in ' need to be addressed explicitly to strengthen the manuscript.

Reviewer #2 (Remarks to the Author):

In this report, the authors provide extremely valuable structural insight into the regulation of signaling activities of CARD9 and CARD11, critical signaling hubs in innate and adaptive immunity. The studies are important and will certainly be of wide interest to the immunology community and those focused on mechanisms of signal transduction in general.

The study will be significantly enhanced if the authors would address the following issues:

Major points:

1) All NF- κ B activation assays of the various expression constructs should be evaluating the proteins' activity at equivalent expression levels when comparing relative signaling activity. These proteins do not show a linear relationship between expression level and signaling output in these assays, so showing the activity resulting from non-equivalent expression levels can be very misleading. Fig 2C and Fig 3B should be repeated to fix this issue.

2) The DAmFRET studies should be performed in mammalian cells at physiological concentrations similar to levels of expression under which native CARD9 and CARD11 signaling occurs. An important issue in the field is that the studies of CARD9 and CARD11 polymerization, and their ability to promote Bcl10 polymerization into filaments, have been done under conditions of high overexpression. There has been little evidence that CARD-protein filaments and Bcl10 filaments appreciably occur in cells under native conditions of signaling, for example in dendritic cells downstream of Dectin triggering, or in B and T lymphocytes downstream of TCR and BCR triggering. The authors DAmFRET assay would provide valuable insight into this if they can perform it in a physiologically relevant context with native, full-length CARD9 and CARD11 proteins downstream of native receptor triggering. The authors state that they used yeast for this because other more relevant cells have other death-domain containing proteins. The presence of these other death-domain containing proteins should not interfere with the results in mammalian cells if the authors' conclusions are correct.

3) The authors discuss the promotion of CARD polymerization and Bcl10 polymerization as an output of signaling without discussing Bcl10 polyubiquitination, or how polymerization can impact Bcl10 polyubiquitination. For example linear ubiquitination of Bcl10 by the LUBAC complex, induced by CARD11 has been shown to determine CARD11 signaling output, and yet it is not clear that Bcl10 polymerization would even be possible with Bcl10 ubiquitylated. Perhaps Bcl10 filaments are just an artifact of overexpression? Or just an artifact of using fragments of CARD proteins that do not include the majority of the proteins' domains that have been shown to be required for signaling (like the rest of the coiled-coil and the MAGUK domains in CARD11)? The authors should discuss these issues explicitly without ignoring what has been previously published in the field.

Minor Point:

In several points the authors refer to a CARD-coiled-coil interface that they have discovered. It's not just the CARD: coiled-coil interface that promotes autoinhibition and that is involved in intramolecular binding, isn't the region in between the CARD and coiled-coil (α -linker) involved? The authors should use the CARD-linker-coiled-coil interface term uniformly in the manuscript. Perhaps the authors should use the previous term for this region that is already in the literature – the LATCH domain, which has been functionally defined. What structural elements observed in this study make up the previously described LATCH domain? If this were stated explicitly, it would help readers familiar with the previous literature.

Reviewer #3 (Remarks to the Author):

This manuscript presents an auto-inhibition mechanism of CARD9/CARD11 in immune cells using different methodologies e.g. structure determination via NMR, HD exchange, FRET in a cellular context, and chimeric protein construction. The conclusion made by the authors is clear and written in a proper way. The novelty of the work lies in the identification of the interface between CARD and coil-coil, linker/ID domains and correlate this large interfaces with many disease related mutants and post-transnational modification. The auto-inhibition mechanism of CARD9 was extensively investigated and presented in a solid way, however, the study of auto inhibition mechanism of CARD11 is weak. The authors showed the TROSY spectrum of CARD11(8-172) and claimed CARD11 was in a molten globular state probably due to heterogeneous conformations or high-order self associations. This could be further clarified by DLS measurement or others biophysical measurements. Moreover, the NMR spectra of mutants that disrupt the interface between CARD and coil coil region of CARD11 will also be interesting to reveal how these mutants affect the conformation(s) of CARD11 (8-172).

The author mentioned that CARD alone of CARD11 exhibits a well dispersed spectrum indicating a single conformation and a well folded state. As the CARD itself did not form high order oligomers, it is a bit confusing how the auto-inhibition works in the case of CARD11? The author mentioned the ID domain of CARD11 is important for the auto-inhibition. We would also like to see how is the NMR spectra of CARD 11 (8-178) looks in the presence of ID domain.

Reviewer #4 (Remarks to the Author):

This reviewer was specifically advised to comment on the rigor of the HDX-MS experiments. The HDX-MS data were acquired using an HDX robotic system coupled to a Q-Exactive (60K resolution, m/z 200). Passed on the error bar provided in the relevant figures the exchange data seem highly reproducible. The authors report the HDX-MS data prior and after correction using a previously published method by one of the co-authors (by B.T. Walter). Despite that the HDX-data are undoubtedly being of high quality the reviewer would like inviting the authors to provide statistic evaluations of the experimental HDX data. In addition, the supplemental materials would be a good place for detailed reporting of the peptide-level HDX-MS data before and after corrections to further allow the readers to follow in more detail the calculations of protection factors.

We thank the reviewers for taking the time to review our manuscript and for their thoughtful comments and suggestions. We have responded point-by-point to each comment and modified the manuscript where applicable. Large changes in the text have been copied as italics below the corresponding reviewer comment. We have included a revised manuscript file as well as a file highlighting all changes from our original submission.

Reviewer #1:

1. The authors need to address the following concerns in this part. For Figure 5B. Firstly, the author mentioned a sharp increase in polymerization at concentration above 100uM, however the inset in figure 5B was determined by vertical dash line at approximately 50uM. If the line is to be adjusted to 100uM (also why figure S5D is cited here? Maybe referred to figure S4D?). The difference between CARD91-142/107E-mEos3 and CARD91-142/WT-mEos3 does not seem significant when gated at 100uM. Additional statistical analysis would be helpful to clarify the statement here. Are the chimeric constructs used in Figure 3 (luciferase assay) behave similarly if using this fluorescent assay or using FP? Comparing the same constructs in different assays would build a strong linkage between biochemical and cellular assays.

- We thank the reviewer for their examination of the data and comments, and apologize for the lack of clarity in exactly what is being plotted in the inset. In Figure 5B, the left plot shows CARD9¹⁻⁹⁷ polymerization occurring at concentrations above ~100 μM. The inset in Fig 5B shows the FRET levels for each of the three proteins binned between 50-60 μM. In this concentration regime, CARD9¹⁻⁹⁷ remains predominantly monomeric, while the FRET level for CARD9^{1-142/WT} is elevated slightly due to dimerization. Due to disruption of the CARD-coiled-coil interface, CARD9^{1-142/107E} exhibits significantly elevated FRET levels above both CARD9¹⁻⁹⁷ and CARD9^{1-142/WT} in this regime corresponding to polymerization. Measuring the FRET levels instead in the 100-200 μM regime indeed shows no difference between CARD9¹⁻⁹⁷ and CARD9^{1-142/107E} due to polymerization of both constructs at these higher concentrations, while the CARD9^{1-142/WT} FRET level remained essentially unchanged relative to the 50-60 μM binning. To clarify these comparisons, we have added log-scale tic-marks to the graphs in Figure 5B and explicitly stated in the figure legend what binning was used to generate the inset figure. As discussed below, we now also explicitly compare the FRET levels binned between 100-200 μM in Supplementary Figure S3.
- The reviewer is correct that the citation should have been to S4D. This error has been corrected in the manuscript, although due to other changes, the figure is now S3C.
- We have now included as Figure S3B data binned at 50-60 μM and 100-200 μM generated from three replicate experiments of those depicted in Figure 5B, along with statistical comparisons of the mean FRET levels of each binning. These results corroborate the qualitative assessment of the data described above and in the manuscript. Specifically, at 50-60 μM, the increase in FRET of CARD9^{1-142/WT} and CARD9^{1-142/107E} above CARD9¹⁻⁹⁷ are found to be significant, as is the increase in FRET of CARD9^{1-142/107E} above CARD9^{1-142/WT}. At 100-200 μM binning, the increase in FRET from both CARD9^{1-142/107E} and CARD9¹⁻⁹⁷ as compared to CARD9^{1-142/WT} is found to be significant, while the FRET levels of CARD9^{1-142/107E} and CARD9¹⁻⁹⁷ are not found to exhibit a significant difference.
- We appreciate the reviewer's impulse to utilize the most biologically relevant protein constructs across assays. In our hands, however, isolated protein comprising the longer CARD9 and CARD11 constructs are poorly behaved, and are thus incompatible with our biochemical assays. We nonetheless regard our results as having built a strong linkage between the biochemical and cellular assays. As is common in structural studies of large multi-domain proteins (including CARD11¹), we have translated structurally and biochemically based hypotheses based on isolated

regions of the protein which are compatible with *in vitro* assays for testing in the full-length protein in cellular assays; all results from both *in vitro* and cellular assays are consistent with our proposed models of autoinhibition and activation.

2. The procedures leading to the identification of the helical symmetry could be better elaborated, perhaps with the facilitation of power spectrum map. Interestingly, the helical symmetry (5Å rise, and -101 degree rotation) matched the Bcl10 helical symmetry perfectly, as well as all the other existing CARD filaments (MAVS-CARD, ASC-CARD, pro-CASP1-CARD, NLRC4-CARD, RIP2-CARD). In addition, this manuscript is also another good example of showing how different upstream CARDS could activate the shared common downstream adaptors, like in the cases of RIG-I/MDA5-MAVS, NOD1/2-RIP2. A insightful addition to the existing literature.

- As described in our Experimental Methods, our initial helical symmetry used was previously determined from a NS-EM reconstruction of the CARD filaments. We have elaborated this description to include the specific method utilized previously, the Spring routine Segclassreconstruct. These parameters were optimized using the helical processing module of RELION. We have also now included in the Experimental Methods the range of helical parameters that were tested in this optimization (4.8-5.4 Å rise and a 100°-104° rotation). Finally, we have also included in Figure S5D a representative power spectrum of one of the 2D class averages generated during the refinement with the prominent 1-, 3-, 4- and 7- start layer lines labeled.
- The reviewer is correct in that the CARD9 CARD structure joins a growing body of literature indicating that CARD filaments adopt a surprisingly narrow range of helical parameters. We have included in the manuscript a sentence highlighting this fact: *The CARD adopts the canonical helical architecture of a death domain, with helical parameters of a 5.11 Å rise and 101.6° left-handed rotation, consistent with a growing body of literature demonstrating a narrow range of parameters adopted by CARDS (4.85-5.13 Å rise and a 100.2°-101.6° left-handed rotation) including that of Bcl10.*

3. However, in order to extend their conclusions to CARD9/11-Bcl10, the authors need to demonstrate that these two complexes are indeed similar. This is particularly important to claim CARD11 signaling mechanism in this study, since its structure is still unknown. Maybe they can consider building models of the potential ways how CARD9/11 could interact with Bcl10, or how CARD11 filament should look like (using homologous models), and perform mutagenesis like what they did in CARD9, then using negative stain EM or luciferase assays to test these hypothesis.

- Given the currently unknown structure of the CARD11 CARD helical assembly, the reviewer is correct that we have an opportunity to also generate a model of the activated CARD11 assembly. Given the highly similar structure of the monomeric CARDS of CARD9 and CARD11², and the overall conservation in the helical architecture among all CARD filaments noted by the reviewer, we anticipate that the CARD11 and CARD9 helical structures will be highly homologous. To generate such a CARD11 filament model, we used Rosetta to thread the CARD11 sequence into the CARD9 CARD filament structure and allowed for relaxation into a local minimum. Based on this model, we generated a number of CARD11 CARD constructs containing single site mutations in residues engaged in inter-CARD interactions in the model. Like the CARD9 CARD, the CARD11 CARD is capable of filament formation upon concentration *in vitro*. We thus incubated each mutant at a concentration double that required for the WT CARD11 CARD to form filaments and found that in all cases, the mutations blocked filament formation as monitored by NS-EM. The mutated sites comprise at least one residue from each of the three canonical CARD-CARD interaction interfaces. A description of this modeling, mutagenesis, and polymerization assay have been added to the manuscript. The CARD11 filament model has likewise been added in Figure S7, along with NS-EM micrographs demonstrating disruption of

CARD11 CARD polymerization by the interface mutants. *Given the structural similarity of the monomeric CARD9 and CARD11 CARDS⁴², we anticipated that the CARD11 CARD (CARD11⁸⁻¹⁰⁹) forms a helical assembly in a similar manner. To probe the CARD11 CARD helical assembly, we used Rosetta to generate a model of the CARD11⁸⁻¹⁰⁹ filament based on our EM structure of the CARD9 CARD filament. As shown in Figure S7A, our model predicts utilization of the same three canonical interfaces; while the residues involved in the interactions are not strictly conserved with those of CARD9, the interaction remains predominantly electrostatic. To experimentally validate our model, we generated mutations within each of the interfaces at residues involved in putative inter-CARD interactions. Unlike the WT CARD11 CARD, which forms filaments in vitro at 250 μ M, each of the mutants were incapable of filament formation at double that concentration (Figure S7B), confirming their relevance in mediating the CARD11 CARD helical assembly.*

Reviewer #2:

1. All NF- κ B activation assays of the various expression constructs should be evaluating the proteins' activity at equivalent expression levels when comparing relative signaling activity. These proteins do not show a linear relationship between expression level and signaling output in these assays, so showing the activity resulting from non-equivalent expression levels can be very misleading. Fig 2C and Fig 3B should be repeated to fix this issue.

- We agree with the reviewer that, ideally, we would present all signaling data using equivalent levels of protein. Following the reviewer's suggestion, we titrated levels of transfected plasmid for the four destabilizing mutants studied in Figure 2C (C37Y, E81D, L85Y, and Y86F); unfortunately, we found that even upon transfection with much higher plasmid concentrations, the expressed levels of these four CARD9 mutants were minimally increased and remained much lower than for WT CARD9. For L85Y, which we consistently observe elevated NF- κ B activation even at these lower protein concentrations, we are nonetheless able to conclude that the mutation is activating. For the other three mutations, we are only able to conclude that the mutations, which are activating in CARD11, are destabilizing in CARD9. Rather than exclude these data from the manuscript, we have instead opted to continue to include the results from the entire panel that we designed and explicitly describe in the text that we were unable to draw conclusions about the activation potential of these particular mutants. We have modified the manuscript text to explain that we were unable to substantially increase protein levels of these constructs and to clarify that we cannot make conclusions about the activating potential of C37Y, E81D, and Y86F. *Four of the mutants tested (C37Y, E81D, L85Y, and Y86F) appear to destabilize CARD9, as indicated by significantly lower protein expression; increasing the concentration of transfected plasmid had a minimal effect on protein levels of these mutants. For L85Y, we observed consistently higher NF- κ B activation despite the lowered protein expression, confirming that the L85Y mutation is activating. Because we were unable to match WT protein expression for C37Y, E81D, and Y86F, we were unable to determine conclusively whether these mutations activate CARD9 as they do CARD11.*
- For the constructs depicted in Figure 3, we were able to modulate protein concentration effectively by titrating levels of transfected plasmid, and have thus repeated the experiments while utilizing a range of protein concentrations for each construct, allowing for direct comparison of the signaling output at comparable protein concentrations between constructs.

2. The DAMFRET studies should be performed in mammalian cells at physiological concentrations similar to levels of expression under which native CARD9 and CARD11 signaling occurs. An important issue in the field is that the studies of CARD9 and CARD11 polymerization, and their ability to promote

Bcl10 polymerization into filaments, have been done under conditions of high overexpression. There has been little evidence that CARD-protein filaments and Bcl10 filaments appreciably occur in cells under native conditions of signaling, for example in dendritic cells downstream of Dectin triggering, or in B and T lymphocytes downstream of TCR and BCR triggering. The authors DAMFRET assay would provide valuable insight into this if they can perform it in a physiologically relevant context with native, full-length CARD9 and CARD11 proteins downstream of native receptor triggering. The authors state that they used yeast for this because other more relevant cells have other death-domain containing proteins. The presence of these other death-domain containing proteins should not interfere with the results in mammalian cells if the authors' conclusions are correct.

- We thank the reviewer for this comment. We agree that application of the DamFRET approach in a mammalian system would potentially provide an avenue to address important open questions in the field. Development of such a system would, however, require substantial technical development that is well beyond the scope of this study; likewise, were such a system developed, it would be best communicated in a separate manuscript in which the details of the method development as well as any caveats could be more thoroughly addressed. For the targeted biophysical questions being addressed in this manuscript, the DamFRET assay in yeast provides an ideal system to monitor concentration-dependent polymerization and nucleation. Our conclusions in no way address the relevance of other potential death domain interactions of either the CARD9 or Bcl10 CARDS; CARD9 in particular has been shown to interact with a number of other CARD containing proteins (e.g. NOD2³, RIPK1⁴, and MAVS⁵) through poorly defined mechanisms; these potential interactions or inherent mechanisms to regulate CARD9 or Bcl10 polymerization in native cells could, in fact, obscure the inherent assembly properties of the CARD9 and Bcl10 CARDS. Devoid of these complicating factors, the yeast DAMFRET system has allowed us to quantitatively monitor the impact of the CARD-coiled-coil interface on concentration-dependent polymerization in the context of the eukaryotic cytosol; as well as, for the first time, demonstrate that Bcl10 exhibits a significant energetic barrier to polymerization and that CARD9 CARD templates can effect Bcl10 polymerization across all concentrations tested, a prerequisite for CARD9/11 nucleation to act as an initiating signal. We acknowledge that the manuscript as currently written suggests that the use of yeast for the assay was a choice due to the lack of other death domains, while, in fact, the assay has only been established in yeast, and the absence of complicating factors such as other death domains is a built-in advantage of such a system. We have reworded this section to more accurately reflect our use of the DAMFRET system in this study. *The DAMFRET assay has been established in the budding yeast, S. cerevisiae, which has the advantage of not containing native death domain proteins or associated regulatory mechanisms that could potentially obscure intrinsic CARD assembly properties.*

3. The authors discuss the promotion of CARD polymerization and Bcl10 polymerization as an output of signaling without discussing Bcl10 polyubiquitination, or how polymerization can impact Bcl10 polyubiquitination. For example linear ubiquitination of Bcl10 by the LUBAC complex, induced by CARD11 has been shown to determine CARD11 signaling output, and yet it is not clear that Bcl10 polymerization would even be possible with Bcl10 ubiquitylated. Perhaps Bcl10 filaments are just an artifact of overexpression? Or just an artifact of using fragments of CARD proteins that do not include the majority of the proteins' domains that have been shown to be required for signaling (like the rest of the coiled-coil and the MAGUK domains in CARD11)? The authors should discuss these issues explicitly without ignoring what has been previously published in the field.

- The reviewer is correct in that there are a range of reports in the literature as to the critical downstream steps involved in activation of NF- κ B and other transcription factors after activation of CARD9 or CARD11. These include the study referenced by the reviewer in which LUBAC-mediated linear ubiquitination serves to modulate the degree of signaling output⁶, as well as other studies which suggest that the LUBAC enzymatic activity is dispensable for NF- κ B activation⁷.

The extent to which these conflicting studies reflect system specific, cell-type specific, or stimulus specific differences in CARD11-initiated signaling have yet to be clarified. Further, the extent to which these findings regarding the critical outcomes downstream of CARD11 activation are reflected in CARD9 activation have yet to be determined. As our study was focused primarily on revealing the structural mechanisms of autoinhibition and activation of CARD9 and CARD11, rather than clarifying the specific (and likely isotype-specific, cell-type-specific, and stimulation-specific) downstream impacts of that activation, we decided to describe the signaling outcomes relatively broadly and generally.

- We suggest that the importance of polymerization to Bcl10-mediated signaling and the relevance of linear ubiquitination to Bcl10-mediated signaling need not be mutually exclusive, even given that ubiquitination within a Bcl10 polymer would likely be sterically excluded. Specifically, it has been demonstrated that signaling puncta that form downstream of TCR activation contain both stable and rapidly equilibrating populations of Bcl10⁸. Bcl10-polymerization-mediated concentration of binding partners may play a role to activate ubiquitin ligases and kinases while filament-terminal and/or labile populations of Bcl10 could simultaneously be linearly ubiquitinated to enhance signaling. That NF- κ B activation can be induced directly through Bcl10 overexpression, and that polymerization-blocking, single-point mutation along multiple interfaces is sufficient to block this activation indicates that Bcl10 polymerization alone is capable of inducing signaling¹. Further, that Bcl10 mutants that prevent Bcl10 polymerization can act in a dominant-negative fashion to block CARD11-induced NF- κ B activation suggests that Bcl10 polymerization remains important for signaling downstream of CARD11 activation¹. Collectively, these findings and the findings of our study suggest that at least one mechanism by which the CARD11, CARD9, and Bcl10 CARDS function is via the formation of helical assemblies, consistent with the known activation mechanisms of many other death domains^{9,10}.
- While we continue to focus the manuscript on describing the autoinhibition and activation mechanisms of CARD9 and CARD11, we agree with the reviewer that it is important to present a full picture of the previously published literature surrounding the CARD9/11-Bcl10 signaling axis. We have thus included additional language in the introduction describing in more detail the downstream outcomes of CARD9/11 activation, including linear Bcl10 ubiquitination, thought to be important for signaling. *The CARDS of CARD9 or CARD11 are thought to form a nucleating helical template that promotes polymerization of Bcl10 via the Bcl10 CARD, which then recruits downstream factors including MALT1, ciAPs, TRAF6, and the LUBAC complex that mediate subsequent ubiquitination of multiple members of the complex, including both linear and K63-linked polyubiquitination of Bcl10^{1,6,11}. These ubiquitination events ultimately lead to activation of IKK and subsequent phosphorylation, ubiquitination, and degradation of I κ B, thereby releasing NF- κ B to translocate to the nucleus and induce transcription.*

4. In several points the authors refer to a CARD-coiled-coil interface that they have discovered. It's not just the CARD: coiled-coil interface that promotes autoinhibition and that is involved in intramolecular binding, isn't the region in between the CARD and coiled-coil (α -linker) involved? The authors should use the CARD-linker-coiled-coil interface term uniformly in the manuscript. Perhaps the authors should use the previous term for this region that is already in the literature – the LATCH domain, which has been functionally defined. What structural elements observed in this study make up the previously described LATCH domain? If this were stated explicitly, it would help readers familiar with the previous literature.

- The reviewer is correct that the linker between the CARD and coiled-coil is critical for maintaining the closed conformation, as many of the sites of hyperactivating mutations in both CARD9 and CARD11 exist in this region (e.g. F103 and I107 in CARD9). We have opted to describe the interface as the 'CARD-coiled-coil' interface for the sake of brevity; however, we now explicitly state upon introduction of the interface that the term 'CARD-coiled-coil' interface refers to the entire CARD-linker-coiled-coil interface. We considered the use of the 'LATCH

domain' as a substitute for 'linker-coiled-coil', however decided against it for two reasons: first, given the new structural insight provided by this study, it is apparent that the previously described LATCH domain is not a structural domain, but rather comprises a linker and a short segment of a longer coiled coil domain; second, the interface between the CARD and the coiled-coil extends significantly beyond the previously described LATCH. It remains to be determined to what extent these interactions beyond the previously described LATCH are critical for maintenance of the interface (the work of Chan, et al. (2013) may suggest that they are not critical in CARD11), however, there are clear interactions between the CARD9 CARD and coiled-coil residues up to Q127 (Figure 1F), while the homologous region of the LATCH as previously described only extends to residue 118. The work of Chan, et al. was certainly vital to understanding the functional importance of the CARD-coiled-coil interface in our structure; in order to orient readers more easily relative to this previous literature, we have included, in Figure S1F, a mapping of the LATCH onto the CARD9⁹²⁻¹⁴² structure in orientations that can be readily compared to Figures 1E and 1F.

Reviewer #3:

1. The authors showed the TROSY spectrum of CARD11(8-172) and claimed CARD11 was in a molten globular state probably due to heterogeneous conformations or high-order self associations. This could be further clarified by DLS measurement or others biophysical measurements. Moreover, the NMR spectra of mutants that disrupt the interface between CARD and coil coil region of CARD11 will also be interesting to reveal how these mutants affect the conformation(s) of CARD11 (8-172).

- We thank the reviewer for the suggestion to more completely characterize the CARD11⁸⁻¹⁷² construct. We have analyzed the construct by SEC-MALS, which indicates a monodisperse population with a molecular weight of 36.5 kDa, consistent with the expected 39.4 kDa mass of a dimer. As suggested, we also measured CARD11⁸⁻¹⁷² by DLS, both at 10 μ M and 100 μ M; at 10 μ M, the polydispersity index (PDI) was 18%, indicating a largely monodisperse species, while at 100 μ M, the PDI was 40%, indicating some degree of polydispersity associated with self-association. Consistent with this analysis, when comparing ¹⁵N-TROSY spectra of CARD11⁸⁻¹⁷² at 50 μ M and 100 μ M (with the 50 μ M experiment collected using 4-times the number of scans to generate equivalent signal-to-noise), we observed a modest improvement in the quality of the spectrum at 50 μ M, indicating some degree of self-association at these concentrations. We conclude, then, that the poor spectral quality of the CARD11⁸⁻¹⁷² dimer stems both from internal conformational dynamics and from self-association, which is likely due to the increased accessibility of the CARD due to the relatively weak CARD-coiled-coil interface. We have included a description of this biophysical characterization of CARD11⁸⁻¹⁷², along with ¹⁵N-TROSY spectra of CARD11⁸⁻¹⁷² at 50 μ M and 100 μ M in Figure S1A. *While SEC-MALS analysis of CARD11⁸⁻¹⁷² indicated a molecular weight of 36.5 kDa, consistent with an expected dimeric molecular weight of 39.4 kDa, dynamic light scattering analysis at 100 μ M revealed a polydispersity index of 0.40, indicating some degree of self-association, which was corroborated by a slight improvement in the NMR spectral quality upon 2-fold dilution (Figure S1A).*
- As suggested, we generated CARD11⁸⁻¹⁷² constructs containing activating mutations (V119E and G126D) and collected NMR spectra. As is observed for CARD9²⁻¹⁵² (Figures 4D and S2D), we also observe a further degradation of the CARD11⁸⁻¹⁷² spectral quality upon disruption of the interface, consistent with the model that these mutations further disrupt the interface and further increase conformational flexibility in the protein. We have included these spectra below.

2. The author mentioned that CARD alone of CARD11 exhibits a well dispersed spectrum indicating a single conformation and a well folded state. As the CARD itself did not form high order oligomers, it is a

bit confusing how the auto-inhibition works in the case of CARD11? The author mentioned the ID domain of CARD11 is important for the auto-inhibition. We would also like to see how is the NMR spectra of CARD 11 (8-178) looks in the presence of ID domain.

- While it has not been previously reported, upon concentration *in vitro*, the CARD11 CARD will form filaments comparable to those formed by the CARD9 CARD. In conjunction with our response to a suggestion by Reviewer #1, we have now included NS-EM micrographs demonstrating that the WT CARD11 CARD is able to form filaments, while disruption of the CARD-CARD interfaces blocks these filaments from forming (Figure S7B). In isolation, both the CARD9 and CARD11 CARDS only form these filaments at high concentration (100s of μM), and the filament formation can be inhibited both by increasing salt concentration, due to the predominantly electrostatic nature of the interactions, or by maintaining the CARDS at an elevated temperature. By maintaining low protein concentration, elevated salt concentration, and elevated temperature, both the CARD9 and CARD11 CARDS can be kept in a monomeric state, allowing for collection of well dispersed NMR spectra. We suggest that the CARDS alone would be unable to polymerize under physiological conditions, but, as has been shown for multiple other death domains, additional interactions by other domains in the protein (the larger coiled-coil domain in the case of CARD9 and CARD11) facilitate bringing the death domains together to a high local concentration that allows for helical assembly.
- We have collected NMR spectra of ^{15}N -CARD11⁸⁻¹⁷² in the presence of unlabeled CARD11 ID (CARD11⁴⁴¹⁻⁶⁷¹) as well as ^{15}N -CARD11⁴⁴¹⁻⁶⁷¹ in the presence of unlabeled CARD11⁸⁻¹⁷². For CARD11⁸⁻¹⁷², we observe the line-broadening of a handful of weak peaks, indicating an interaction, but no significant improvement in the spectrum. This is not unexpected, as the ID has been shown to exhibit multivalent interactions with the CARD, larger coiled-coil domain, and additional regions of CARD11 using a number of redundant elements¹², and likely requires these numerous interactions to maintain the CARD in a fully closed state. When observing the ID, we note that the domain is predominantly unstructured, but that a number of peaks exhibit significant shifts and line-broadening in the presence of CARD11⁸⁻¹⁷², again confirming that the two domains do, in fact, interact. We have included these spectra below.

Reviewer #4:

1. This reviewer was specifically advised to comment on the rigor of the HDX-MS experiments. The HDX-MS data were acquired using an HDX robotic system coupled to a Q-Exactive (60K resolution, m/z 200). Passed on the error bar provided in the relevant figures the exchange data seem highly reproducible. The authors report the HDX-MS data prior and after correction using a previously published method by one of the co-authors (by B.T. Walter). Despite that the HDX-data are undoubtedly being of high quality the reviewer would like inviting the authors to provide statistic evaluations of the experimental HDX data. In addition, the supplemental materials would be a good place for detailed reporting of the peptide-level HDX-MS data before and after corrections to further allow the readers to follow in more detail the calculations of protection factors.

- Thank you to this reviewer for the insightful comments. We have honored the request for additional information through the addition of two supplementary figures that show all peptides included in the manuscript along with excel tables listing protection factors (PFs) and PF range for each peptide consistent with the replicate measurements for the purpose of statistical evaluation. These figures and tables are included as part of the Source Data excel document which we have now included. The PF range required development of a new method to translate variance in deuterium uptake to variance in protection factor (described briefly in the Source Data text). Readers interested in following protection factor calculations more closely will appreciate the additional data in SI figures and reference to a graph theoretic method that empirically defines

PFs described by BT Walters previously (*Anal Chem*, 2017, 89(2): p. 1049-1053.). Thank you again for your time to review the paper, we hope this has provided appropriate clarification.

References

1. Qiao, Q. et al. Structural architecture of the CARMA1/Bcl10/MALT1 signalosome: nucleation-induced filamentous assembly. *Mol Cell* **51**, 766-79 (2013).
2. Holliday, M.J. et al. Picomolar zinc binding modulates formation of Bcl10-nucleating assemblies of the caspase recruitment domain (CARD) of CARD9. *J Biol Chem* **293**, 16803-16817 (2018).
3. Hsu, Y.M. et al. The adaptor protein CARD9 is required for innate immune responses to intracellular pathogens. *Nat Immunol* **8**, 198-205 (2007).
4. Cao, M. et al. Dectin-1-induced RIPK1 and RIPK3 activation protects host against *Candida albicans* infection. *Cell Death Differ* (2019).
5. Poeck, H. et al. Recognition of RNA virus by RIG-I results in activation of CARD9 and inflammasome signaling for interleukin 1 beta production. *Nat Immunol* **11**, 63-9 (2010).
6. Yang, Y.K. et al. Molecular Determinants of Scaffold-induced Linear Ubiquitylation of B Cell Lymphoma/Leukemia 10 (Bcl10) during T Cell Receptor and Oncogenic Caspase Recruitment Domain-containing Protein 11 (CARD11) Signaling. *J Biol Chem* **291**, 25921-25936 (2016).
7. Dubois, S.M. et al. A catalytic-independent role for the LUBAC in NF-kappaB activation upon antigen receptor engagement and in lymphoma cells. *Blood* **123**, 2199-203 (2014).
8. Rossman, J.S. et al. POLKADOTS are foci of functional interactions in T-Cell receptor-mediated signaling to NF-kappaB. *Mol Biol Cell* **17**, 2166-76 (2006).
9. Ferrao, R. & Wu, H. Helical assembly in the death domain (DD) superfamily. *Curr Opin Struct Biol* **22**, 241-7 (2012).
10. Wu, B. et al. Molecular imprinting as a signal-activation mechanism of the viral RNA sensor RIG-I. *Mol Cell* **55**, 511-23 (2014).
11. Wu, C.J. & Ashwell, J.D. NEMO recognition of ubiquitinated Bcl10 is required for T cell receptor-mediated NF-kappaB activation. *Proc Natl Acad Sci U S A* **105**, 3023-8 (2008).
12. Jattani, R.P., Tritapoe, J.M. & Pomerantz, J.L. Cooperative Control of Caspase Recruitment Domain-containing Protein 11 (CARD11) Signaling by an Unusual Array of Redundant Repressive Elements. *J Biol Chem* **291**, 8324-36 (2016).

REVIEWERS' COMMENTS:

Reviewer #1 (Remarks to the Author):

Reviewer Comments:

1. The authors of the manuscript have taken significant effort to address our comments in the previous round. In particular, they specifically conducted further investigations and produced new data/analysis to support their claims. In Figure 5 and Figure S3, new analysis were included to better supported their claims compared to the previous draft. They have also included detailed methods solving the helical structure of CARD9 (Figure S5) and computer modeling of CARD11 (Figure S7). With the addition of these efforts, the manuscripts is now a solid piece of biochemistry work characterizing the monomer to oligomer transition of CARD9/11, as well as their filament structure.

2. As other reviewers pointed out, applying these structural/biochemical findings into understanding of CARD9/11 cellular activities would provide critical information for fellow cell biologists to understand how CARD9/11 mediated signaling work. However, the authors may wish to choose to use more inclusive descriptions of the new structural findings, noting the differences among recombinant structural approaches, cellular FRET experiments and actual macrophage/T cell biology. In particular, the DAMFRET data in Figure 5B (also there is mixed usage of DAMFRET and AmFRET in figures) is still not strong enough to support the physiological relevance of the study, a better description of the cut-off and elaboration of the mathematical methods would be helpful to support their finding. Perhaps, a pair of control DAMFRET experiments on familiar proteins would better explain/justify their method.

Overall, right now, this is a well conducted structural/biochemical study, providing definitive description of the resting state-oligomer transition of CARD9, and potential oligomer organization of CARD11.

Reviewer #2 (Remarks to the Author):

In the revised manuscript, the authors have addressed some of the issues raised in the previous review. However, two items in the text should be corrected to adequately refer to the large body of previous literature on CARD11 signaling that the authors may not be familiar with.

1) In the third paragraph of the introduction the author state that

"The CARDS of CARD9 or CARD11 are thought to form a nucleating helical template that promotes polymerization of Bcl10 via the Bcl10 CARD, which then recruits downstream factors including MALT1, cIAPs, TRAF6, and the LUBAC complex that mediate subsequent ubiquitination of multiple members of the complex, including both linear and K63-linked polyubiquitination of Bcl10¹³⁻¹⁵"

This sentence is not correct. CARD11 contains multiple domains in addition to the CARD and many papers have shown that CARD11 recruits a large collection of proteins through the CARD, Coiled-coil, and PDZ-SH3-GUK domains. It is not the case that CARD11 recruits Bcl10, and then Bcl10 recruits all subsequent proteins, as the manuscript currently states. For example, the HOIP subunit of the LUBAC complex is recruited to CARD11 through the Coiled-coil of CARD11 in a manner that is completely independent of the CARD domain and Bcl10 recruitment. In addition TRAF6 recruitment to CARD11 also requires the Coiled-coil of CARD11. It might be helpful to the authors to read several recent reviews on CARD11 signaling, including Bedsaul et al, 2018, *Frontiers in Immunology*, Volume 9, article 2105 [DOI 10.3389/fimmu.2018.02105], which summarize what is known about mechanisms of CARD11 signaling.

In the previous review it was pointed out that the authors did not adequately describe the role of HOIP-mediated Bcl10 linear ubiquitination in CARD11 signaling to the IKK complex. The authors have added reasonable reference to this work, but they may wish to remind readers that linear-

ubiquitinated Bcl10 is the moiety that a) interacts directly with the IKK γ subunit of the IKK complex to facilitate kinase activation, and that b) the levels of linear ubiquitinated Bcl10 determine signaling output to NF- κ B. The authors are correct in their response to the previous review that it is unclear how the induction of Bcl10 polymerization could be compatible with the formation of linear ubiquitinated Bcl10, but this lack of clarity in the field should be explicitly discussed in the manuscript rather than be omitted in the discussion just because it might appear to detract from the authors' focus on polymerization.

2) Extensive prior work has already characterized the functional role of the LATCH portion of CARD11 and this work should be described and cited explicitly. Previous work has shown that the LATCH is required for CARD11 autoinhibition, that the LATCH allows for ID binding, and that in the unstimulated state, the LATCH prevents Bcl10 binding, the subsequent ubiquitination of Bcl10, and the action of ubiquitinated Bcl10 on the IKK complex (Chan et al MCB 2013, volume 33, page 429). The explicit description of the LATCH and its prior study should be presented upfront in the manuscript at end of the fourth paragraph of the introduction. The authors may not have done so in previous drafts because of the concern that doing so would diminish the novelty of their study --- on the contrary, being explicit at describing the LATCH domain and its prior study would only enhance interest in the structural information they have provided in this extremely valuable study.

Reviewer #3 (Remarks to the Author):

This paper is greatly improved by the authors' revision. My concerns are addressed. I have no further comments.

We thank the reviewers for their additional comments and suggestions. We have responded point-by-point to each comment and modified the manuscript where applicable. Changes in the text have been copied as italics below the corresponding reviewer comment.

Reviewer #1:

1. As other reviewers pointed out, applying these structural/biochemical findings into understanding of CARD9/11 cellular activities would provide critical information for fellow cell biologists to understand how CARD9/11 mediated signaling work. However, the authors may wish to choose to use more inclusive descriptions of the new structural findings, noting the differences among recombinant structural approaches, cellular FRET experiments and actual macrophage/T cell biology. In particular, the DAmFRET data in Figure 5B (also there is mixed usage of DAmFRET and AmFRET in figures) is still not strong enough to support the physiological relevance of the study, a better description of the cut-off and elaboration of the mathematical methods would be helpful to support their finding. Perhaps, a pair of control DAmFRET experiments on familiar proteins would better explain/justify their method.

- We agree with the reviewer in the value of translating our structural and biochemical findings regarding activation of CARD9 and CARD11 into the context of native signaling. However, implementing the DAmFRET assay or other methods to directly monitor CARD assembly in the context of native signaling would require extensive method development beyond the scope of the current study. We likewise agree that the DAmFRET assay alone is insufficient to establish physiological relevance of our findings. Rather, the DAmFRET assays establish, in the context of eukaryotic cytosol, that the coiled-coil is sufficient to block CARD assembly, that disruption of the interface restores the ability of the CARD to assemble, that Bcl10 can exist in a super-saturated, nucleation-limited monomeric state, and that the CARD assemblies are sufficient to nucleate Bcl10 polymerization. The physiological relevance of our findings is addressed in Figures 2 and 3, in which we show that maintenance of the CARD-coiled-coil interface inhibits activation and that disruption of the interface in the full-length proteins is sufficient to activate the signaling pathway. Physiological relevance is additionally established through comparison with previous literature demonstrating that Bcl10 polymerization is sufficient to drive signaling, that Bcl10 assembles upon native CARD11 activation, and that disruption of the CARD coiled-coil interface is sufficient to induce recruitment of Bcl10, promote NF- κ B activation, and drive oncogenesis¹⁻⁴. A thorough description of the DAmFRET assay, including mathematical methods, as well as application of DAmFRET to numerous well-described polymerizing targets are provided in the initial paper describing the DAmFRET assay by Khan et al.⁵.

Reviewer #2:

1. In the third paragraph of the introduction the author state that, "The CARDS of CARD9 or CARD11 are thought to form a nucleating helical template that promotes polymerization of Bcl10 via the Bcl10 CARD, which then recruits downstream factors including MALT1, cIAPs, TRAF6, and the LUBAC complex that mediate subsequent ubiquitination of multiple members of the complex, including both linear and K63-linked polyubiquitination of Bcl10¹³⁻¹⁵." This sentence is not correct. CARD11 contains multiple domains in addition to the CARD and many papers have shown that CARD11 recruits a large collection of proteins through the CARD, Coiled-coil, and PDZ-SH3-GUK domains. It is not the case that CARD11 recruits Bcl10, and then Bcl10 recruits all subsequent proteins, as the manuscript currently states. For example, the HOIP subunit of the LUBAC complex is recruited to CARD11 through the Coiled-coil of CARD11 in a manner that is completely independent of the CARD domain and Bcl10 recruitment. In addition TRAF6 recruitment to CARD11 also requires the Coiled-coil of CARD11. It

might be helpful to the authors to read several recent reviews on CARD11 signaling, including Bedsaul et al, 2018, *Frontiers in Immunology*, Volume 9, article 2105 [DOI 10.3389/fimmu.2018.02105], which summarize what is known about mechanisms of CARD11 signaling.

- We agree with the reviewer that the wording of the sentence quoted incorrectly suggests that all subsequent recruitment of downstream factors relies on Bcl10 polymerization, when, in fact, significant literature exists to suggest that other domains of CARD11 (and, by homology CARD9) participate in these recruitments. We have reworded the sentence to clarify that CARD9/11 activation recruits relevant downstream factors both through recruitment of Bcl10 and through other domains in the activated protein. *The CARDS of CARD9 or CARD11 are thought to form a nucleating helical template that promotes polymerization of Bcl10 via the Bcl10 CARD, which, along with other domains of activated CARD9/11, then recruits downstream factors including MALT1, cIAPs, TRAF6, and the LUBAC complex that mediate subsequent ubiquitination of multiple members of the complex, including both linear and K63-linked polyubiquitination of Bcl10.*

2. In the previous review it was pointed out that the authors did not adequately describe the role of HOIP-mediated Bcl10 linear ubiquitination in CARD11 signaling to the IKK complex. The authors have added reasonable reference to this work, but they may wish to remind readers that linear-ubiquitinated Bcl10 is the moiety that a) interacts directly with the IKK γ subunit of the IKK complex to facilitate kinase activation, and that b) the levels of linear ubiquitinated Bcl10 determine signaling output to NF- κ B. The authors are correct in their response to the previous review that it is unclear how the induction of Bcl10 polymerization could be compatible with the formation of linear ubiquitinated Bcl10, but this lack of clarity in the field should be explicitly discussed in the manuscript rather than be omitted in the discussion just because it might appear to detract from the authors' focus on polymerization.

- There exists significant disagreement in the literature as to the general importance of linear ubiquitination in CARD11 signaling, with studies indicating both that LUBAC mediated linear ubiquitination is directly correlated to the degree of NF- κ B activation⁶ and that LUBAC enzymatic activity is dispensable for NF- κ B activation⁷. The source of this discrepancy has yet to be established, and may reflect system, cell type, or stimulus specific signaling events. Further, linear ubiquitination has not, to our knowledge, been demonstrated in the course of CARD9 signaling. Given that our current study is focused specifically on the structural events involved in activation of CARD9/11 and Bcl10 recruitment, and not on the subsequent downstream signaling events, and, in fact, does not address Bcl10 ubiquitination one way or the other, we have opted not to elaborate in detail about any downstream signaling events in our discussion of the system.

3. Extensive prior work has already characterized the functional role of the LATCH portion of CARD11 and this work should be described and cited explicitly. Previous work has shown that the LATCH is required for CARD11 autoinhibition, that the LATCH allows for ID binding, and that in the unstimulated state, the LATCH prevents Bcl10 binding, the subsequent ubiquitination of Bcl10, and the action of ubiquitinated Bcl10 on the IKK complex (Chan et al *MCB* 2013, volume 33, page 429). The explicit description of the LATCH and its prior study should be presented upfront in the manuscript at end of the fourth paragraph of the introduction. The authors may not have done so in previous drafts because of the concern that doing so would diminish the novelty of their study --- on the contrary, being explicit at describing the LATCH domain and its prior study would only enhance interest in the structural information they have provided in this extremely valuable study.

- We certainly agree that previous studies of the hyperactivating mutations in CARD11, especially that of Chan et al, were critical to our understanding of the activation mechanism of CARD9 and CARD11. We have, in fact, already explicitly referenced the Chan et al. paper at the end of paragraph 4 of the introduction, describing that the work of these authors revealed a large number of activating mutations in the N-terminal region of CARD11. The importance of these mutations

in our understanding is not limited to those within the LATCH, as multiple mutations within the CARD itself also map to the CARD-coiled-coil interface; however, we now explicitly state that many of the mutation sites identified map to a short stretch of residues, termed the 'LATCH', and that this region is critical for maintaining CARD11 in an autoinhibited state. *Among these studies, the N-terminal CARD and coiled-coil domains emerged as mutational 'hot-spots', leading Chan, et al. ⁴ to conduct a high-throughput mutagenesis screen on the N-terminal 140 residues of CARD11, in the context of the full-length protein, in search of hyperactivating CARD11 mutants. This study identified 23 sites with hyperactivating mutations in CARD11, many of which mapped to a short stretch of residues (112-130) termed the 'LATCH,' that was shown to be critical in maintaining CARD11 in an autoinhibited state.*

References

1. Qiao, Q. et al. Structural architecture of the CARMA1/Bcl10/MALT1 signalosome: nucleation-induced filamentous assembly. *Mol Cell* **51**, 766-79 (2013).
2. Rossmann, J.S. et al. POLKADOTS are foci of functional interactions in T-Cell receptor-mediated signaling to NF-kappaB. *Mol Biol Cell* **17**, 2166-76 (2006).
3. Juilland, M. & Thome, M. Role of the CARMA1/BCL10/MALT1 complex in lymphoid malignancies. *Curr Opin Hematol* **23**, 402-9 (2016).
4. Chan, W., Schaffer, T.B. & Pomerantz, J.L. A quantitative signaling screen identifies CARD11 mutations in the CARD and LATCH domains that induce Bcl10 ubiquitination and human lymphoma cell survival. *Mol Cell Biol* **33**, 429-43 (2013).
5. Khan, T. et al. Quantifying Nucleation In Vivo Reveals the Physical Basis of Prion-like Phase Behavior. *Mol Cell* **71**, 155-168 e7 (2018).
6. Yang, Y.K. et al. Molecular Determinants of Scaffold-induced Linear Ubiquitylation of B Cell Lymphoma/Leukemia 10 (Bcl10) during T Cell Receptor and Oncogenic Caspase Recruitment Domain-containing Protein 11 (CARD11) Signaling. *J Biol Chem* **291**, 25921-25936 (2016).
7. Dubois, S.M. et al. A catalytic-independent role for the LUBAC in NF-kappaB activation upon antigen receptor engagement and in lymphoma cells. *Blood* **123**, 2199-203 (2014).